# Early Flood Monitoring and Forecasting System Using a Hybrid Machine Learning-Based Approach

Eleni-Ioanna Koutsovili [1,*], Ourania Tzoraki [1], Nicolaos Theodossiou [2] and George E. Tsekouras [3]

1   Department of Marine Sciences, School of the Environment, University of the Aegean, 81100 Mytilene, Greece; rania.tzoraki@aegean.gr
2   Department of Civil Engineering, Polytechnic School, Aristotle University of Thessaloniki, 54636 Thessaloniki, Greece; niktheod@civil.auth.gr
3   Department of Cultural Technology and Communication, University of the Aegean, 81100 Mytilene, Greece; gtsek@ct.aegean.gr
*   Correspondence: ekoutsovili@aegean.gr

**Abstract:** The occurrence of flash floods in urban catchments within the Mediterranean climate zone has witnessed a substantial rise due to climate change, underscoring the urgent need for early-warning systems. This paper examines the implementation of an early flood monitoring and forecasting system (EMFS) to predict the critical overflow level of a small urban stream on Lesvos Island, Greece, which has a history of severe flash flood incidents requiring rapid response. The system is supported by a network of telemetric stations that measure meteorological and hydrometric parameters in real time, with a time step accuracy of 15 min. The collected data are fed into the physical Hydrologic Engineering Center's Hydrologic Modeling System (HEC-HMS), which simulates the stream's discharge. Considering the HEC-HMS's estimated outflow and other hydro-meteorological parameters, the EMFS uses long short-term memory (LSTM) neural networks to enhance the accuracy of flood prediction. In particular, LSTMs are employed to analyze the real-time data from the telemetric stations and make multi-step predictions of the critical water level. Hydrological time series data are utilized to train and validate the LSTM models for short-term leading times of 15 min, 30 min, 45 min, and 1 h. By combining the predictions obtained by the HEC-HMS with those of the LSTMs, the EMFS can produce accurate flood forecasts. The results indicate that the proposed methodology yields trustworthy behavior in enhancing the overall resilience of the area against flash floods.

**Keywords:** flash floods; real-time monitoring; physical hydrological model; deep artificial neural networks; long short-term memory; multi-step predictions

## 1. Introduction

The influence of climate change and urbanization on river and stream flow patterns is exacerbating the issue of urban flooding, making it a growing concern. Flash floods, which are characterized by rapid and unexpected rises in water levels, can be particularly dangerous in urban areas where the flow of water is frequently constrained by man-made structures. In this context, river regulation projects such as channel straightening and shortening, and riverside footpaths, increase the risk of flooding [1]. The urbanization and development process leads to the transformation of natural watercourses into confined, artificial systems known as urban streams, which alter hydrological regimes and increase the flood risk by being highly sensitive to changes in precipitation patterns. In the Mediterranean climate zone, there is an anticipated rise in the occurrence of floods in the upcoming years, leading to significant alterations in the ecological and hydrological patterns of river basins [2].

The Kalloni river basin on Lesvos Island, Greece, in particular, is characterized by a high risk of flooding, primarily from flash floods, which can be attributed to the expected reduction of soil water and groundwater recharge during summer and the substantial

increase in autumn discharge [3]. The Kalloni river basin is a highly significant area due to its rich biodiversity in NATURA 2000 areas, and being designated as a Special Protection Area and a proposed Greek Site of Community Importance [3,4]. However, past incidences of flooding coupled with a reduced riverbed cross-section and sudden high-intensity rainfall have rendered the region prone to flash floods, which pose a threat to the local infrastructure and community [5,6]. As indicated in the Flood Risk Management Plans of the Aegean Islands Water Department, Kalloni is categorized within the Potentially High Flood Risk Zone, which was identified during the initial assessment of flood risk conducted by the Special Secretariat for Water of the Greek Ministry [7]. The frequent incidences of flooding in the area can be attributed to the combination of sloping soils in the general topography and human activities that have altered the landscape and geomorphology of the region [8]. Due to climate change, the region experiences higher mean annual rainfall and temperatures, with a notable rise in autumn precipitation and variation in long-term average discharges, characterized by an upward trend in autumn and a downward trend in summer [3]. This could potentially decrease soil water and groundwater recharge, heightening the vulnerability to flash floods in the Kalloni river basin region, underscoring the importance of comprehensive water management and strategies for mitigating floods [3].

Floods exhibit complex behavior with high uncertainty due to the impact of precipitation intensity, natural geography, and watershed features, resulting in nonlinear, non-stationary, and stochastic flood process [9,10]. Early and precise flood forecasting in urban areas plays a vital role in providing valuable environmental information for decision-making and minimizing the effects of flood-related damage [11,12]. The prediction of flow in urban flood control over the long term is a challenging issue, as various hydrological and meteorological factors are involved [13]. However, machine learning methods have made notable advancements in capturing the physical flow processes of floods, particularly in the context of short- to medium-term predictions [14]. Therefore, machine learning schemes have been spotlighted as useful tools in short- to medium-term urban flood control and prediction. Another key element of flood prediction is real-time monitoring, which utilizes a variety of tools and techniques to gather data on precipitation, water levels, and other relevant factors. To improve the efficiency of flood monitoring systems and ensure early warning, the Internet of Things (IoT) can be integrated with the monitoring process [13]. IoT relies on fields such as wireless sensor networks, embedded and control systems, and automation to enable real-time monitoring [15]. Previous studies have successfully developed and implemented IoT monitoring systems to gather flood-relevant data, including discharge and water level, in near real time, facilitating the timely detection of flood events and the implementation of emergency measures [16]. Combining real-time monitoring with machine learning-based flood prediction models can lead to the development of early warning systems that accurately identify potential flood hazards, reducing the risk to life and property.

In existing literature, a range of approaches can be found for flood forecasting, encompassing both process-driven and data-driven methodologies [17,18]. Both data-driven machine learning techniques and process-based hydrological models have been extensively employed to cope with several classification and regression problems in the field of hydrological sciences [19]. Physical models, such as the Hydrologic Engineering Center's Hydrologic Modeling System (HEC-HMS), are frequently used to simulate and predict flood events [20–22]. According to Wijayarathne & Coulibaly [23], a discharge forecast experiment using deterministic weather prediction indicated that HEC-HMS models exhibit good performance in forecasting within narrow time intervals ahead and are recommended for operational use. Physical models, also referred to as process-driven models, are based on the principles of classical bucket models and incorporate various processes [24]. While physical models have demonstrated significant capabilities in predicting a wide range of flooding scenarios, their effectiveness is often hindered by the need for multiple hydro-geomorphological monitoring datasets and computationally intensive calculations, making

them less suitable for short-term predictions and requiring substantial time and resources for development [9,25]. Process-based models typically require complex calculations, extensive data on hydrology and meteorology, and a thorough comprehension of runoff mechanisms [26]. The effectiveness of each model is determined by such specific conditions, and various limiting factors can contribute to poor flood prediction outcomes. However, with the advancement of scientific and technological developments [27], remote sensing technology has become a more diverse and promising method of acquiring necessary data [17].

In addition to traditional physical models, data-driven machine learning strategies such as neural networks are increasingly used to analyze time series data for flood forecasting as well as operational flood warning systems due to their ability to identify patterns and trends in historical data by capturing complex functions and replicating the random and uncertain characteristics of input and responses [28–30]. These methods involve the statistical correlation between input and output data, disregarding the underlying physical mechanisms of the hydrological process [31]. They can easily integrate mathematical analysis of time series data and utilize samples to identify statistical or causal relationships among hydrological variables, thus effectively predicting both short-term and long-term events effectively with minimal input [17,32–34].

However, the implementation of neural networks in flood prediction is related to certain limitations as they are sensitive to input variations and struggle to capture the watershed runoff generation process when there is no delayed correlation between target and features variables [35]. To address that issue, delayed precipitation and runoff can be added as additional input [19]. For example, Kim et al. [19] compared the hydrological simulation accuracy of data-driven machine learning models and classical process-based hydrological models. Their results demonstrated that the data-driven models can achieve highly accurate forecasts in high-flow regimes, while the process-based models are more reliable tools in low-flow regimes, implying both models have their respective pros and cons.

Based on the above analysis, it is evident that an integrated flood monitoring and forecasting system (EMFS) that combines data-driven machine learning techniques with hydrological models has the potential to effectively deal with impending flooding events in urban catchments with intermittent flow patterns. This paper discusses such a mechanism in terms of a specific example that focuses on the implementation of an early flood monitoring and forecasting system (EMFS) on Lesvos Island, Greece. The EMFS is designed to perform multi-step prediction of the critical overflow level of a small urban stream, providing valuable information to local authorities and residents to allow for a timely and effective response. The system is supported by a network of telemetric stations and a physical, deterministic, semi-distributed hydrological model. Additionally, machine learning data-driven techniques are employed as tools to analyze, monitor, and accurately predict flood events such as deep artificial neural networks with a long short-term memory (LSTM) architecture. The implementation of such an early warning system enhances the region's overall resilience against future flood disasters in addition to assisting in reducing the consequences of flash floods on the local community.

The remaining sections of the paper are synthesized as follows. Section 2 analytically presents the materials and methods employed. Section 3 provides the simulation results, while the corresponding discussion is carried out in Section 4. Finally, Section 5 concludes the paper.

## 2. Materials and Methods

The proposed framework, illustrated in Figure 1, serves as a comprehensive approach for the implementation of an early flood monitoring and forecasting system (EMFS). This framework encapsulates two primary components. First, there is a focus on the construction and evaluation of the physical hydrological model HEC-HMS, which plays a pivotal role in elaborating hydrological time series data. Second, the framework entails the investigation and multi-step forecasting of critical overflow levels through the utilization of data-driven

models. By integrating these two components, the EMFS framework provides a robust system for monitoring and predicting flood events, improving flood management.

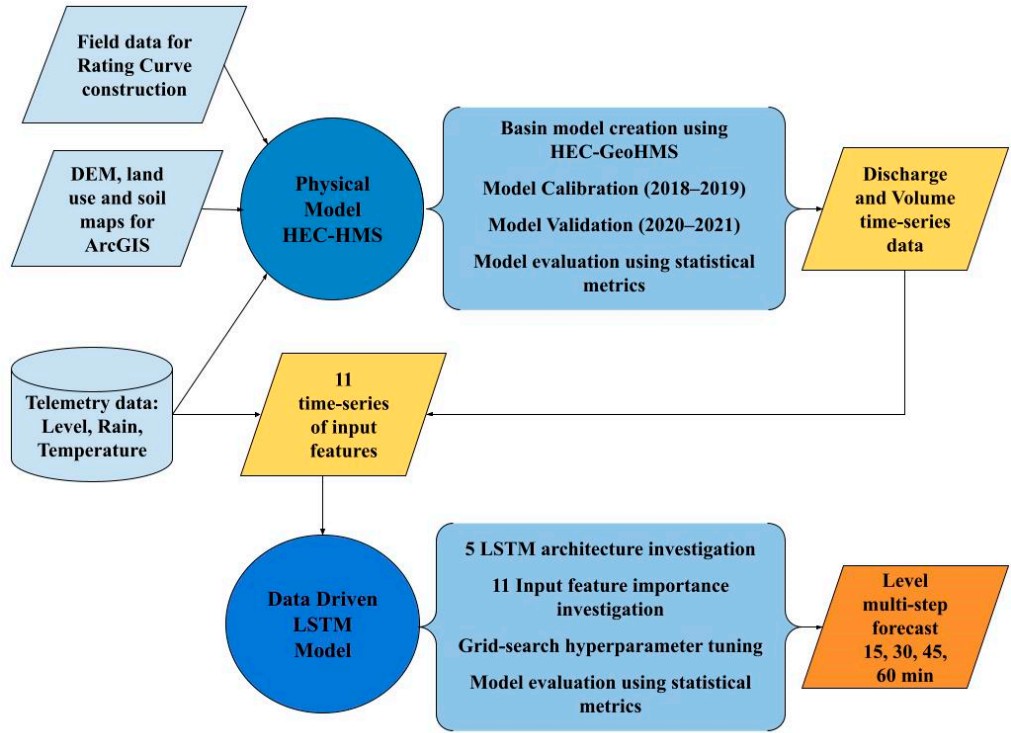

**Figure 1.** Flowchart of methodological framework.

The physical simulation process involves the utilization of a high-resolution digital elevation model (5 m/pixel), land use, and soil maps within the HEC-GeoHMS plugin of ArcGIS. These inputs are employed to delineate, visualize, and extract the basin model and background maps required for conducting the hydrological simulation using the HEC-HMS model. To ensure the model's precision, field measurements are collected for rating curve computation, which is important in model's calibration and validation at a high-frequency time step of 15 min. To support the system, a network of telemetric stations is established, providing real-time meteorological and hydrometric data for both the physical and data-driven models. This data integration enables the incorporation of up-to-date information into the simulation process. The output series produced by the HEC-HMS model, combined with the telemetry time series, form the basis for the 11 groups of investigated input features utilized in the data-driven models. These input features are essential for the accurate prediction and analysis carried out by the data-driven components of the methodology.

This study investigates the adequacy of five LSTM-Network architectures for multivariate time series forecasting, comparing Vanilla LSTM-Networks, Stacked-LSTM Networks, Bidirectional-LSTM Networks, Encoder–Decoder Sequence to Sequence LSTM Networks, and Encoder–Decoder Bidirectional-LSTM Networks, described in detail in Section 2.4. The training phase of each model involves a grid search hyperparameter tuning phase to optimize performance. Additionally, the study explores the impact of different input sequences on various LSTM architectures through a permutation feature importance investigation. During the training process, the LSTM models utilize a time-lag selection of 4 h to make forecasts of water levels with short-term leading times of 15 min, 30 min, 45 min, and 1 h. The performance evaluation encompasses both physical and data-driven models using a range of statistical indicators. The findings contribute to the proposal of a sensitive LSTM architecture with an optimized input sequence, facilitating accurate multi-step prediction of water levels.

## 2.1. Description and Monitoring of the Study Area

The study area under examination, illustrated in Figure 2, is a small urban stream named Kalloni, located on Lesvos Island, Greece. The Kalloni settlement serves as the capital of the Municipality of West Lesvos Island and is the second-largest commercial hub on the North Aegean island. The Municipal Unit of Kalloni has a population of 8420 inhabitants according to the 2021 census [36]. The studied watershed encompasses an area of 40.28 km$^2$ and drains the wider area of the plain, passing through several settlements, ultimately emptying into the Kalloni gulf. The Kalloni river basin encompasses a diverse landscape, ranging from lowlands to mountains, with an average elevation of 300 m. The southern region consists of a large, flat expanse at a similar elevation to sea level, which includes wetlands and marshy areas. The basin predominantly consists of agricultural land, particularly olive groves, with small areas of oak and pine woodlands in the north, brushland habitats in the east, and wetlands and swamps in the south [8]. The study region experiences a Mediterranean climate characterized by an average yearly temperature of around 17 °C and a mean annual rainfall of 514 mm across the Kalloni river basin [3]. The hydrographic network of Kalloni consists of many intermittent streams and has an overall length of 34.92 km.

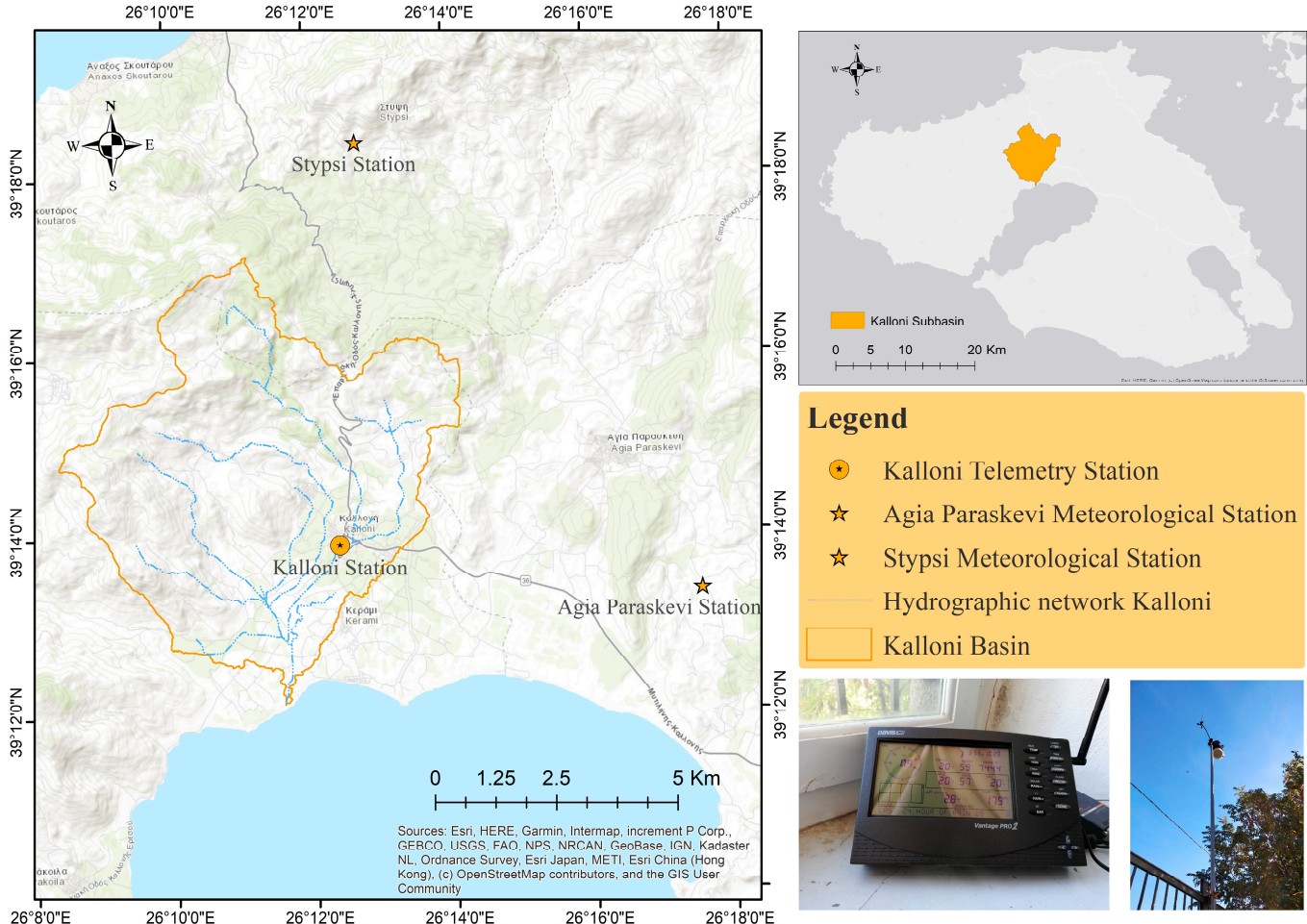

**Figure 2.** Study area and monitoring system location.

A flood monitoring system has been implemented in the Kalloni area, consisting of two meteorological stations and a water-level station that provide real-time telemetry data (Figure 2). The telemetry station in Kalloni has been operating since November 2018, recording the water level in 15-min time steps, while also including a rain gauge. Two additional meteorological stations in Agia Paraskevi and Stypsi have been operational

since 2003 and 2018 respectively, providing a meteorological dataset with high time accuracy. Although the stations have only been operational for a few years, their high time resolution allows for several recordings that can be used for simulation models and close-in-time forecasting with great accuracy, particularly for rapidly rising water levels.

### 2.2. Data Acquisition and Preprocessing

To generate a complete dataset for calibrating and testing the physical hydrological model, various time series covering the period from 2018 to 2022 were created. The telemetry instrument of level recording was used to collect the time series of observed water levels, while the rainfall records were filled with the records from neighboring stations to create a synthetic time series of precipitation. Linear interpolation was employed to address data gaps arising from equipment limitations and human errors during data collection, allowing for the estimation of missing values. Additionally, a 15-min time series of observed discharge was created using the rating curve calculated for the study basin, which was derived from field flow measurements taken in the area from 2019 to 2022 using a portable flow meter device with a propeller. The calculation of the rating curve equation for discharge (Q) and stage (H) was derived using the following formula:

$$Q = 1.796 \times H^{1.02} \tag{1}$$

To ensure compatibility with the HEC-HMS model for optimal forecasting, the 15-min time series data of observed discharge, evapotranspiration, temperature, and precipitation were preprocessed by identifying and cleaning any gaps or irregularities. In addition, the spatial processing and delineation of the sub-basins and hydrographic network of the study catchment were executed in the Arc-GIS environment. This process made use of a high-resolution digital elevation model with a 5-m resolution, along with land use and soil maps. A weighted average method was employed to derive several key hydro-morphological characteristics from these data sources, such as slope, concentration time, and imperviousness. Table 1 describes the raw data sources for the hydrological processing using HEC-GeoHMS 10.4 and HEC-HMS 4.9 software.

**Table 1.** Raw data sources for hydrological modeling.

| Data | Application | Resolution | Source |
|---|---|---|---|
| Water Level | Rating curve construction, HEC-HMS calibration and validation | 15 min | ERMIS-floods platform https://ews.ermis-f.eu/ * |
| Stream flow | Rating curve construction, HEC-HMS calibration, and validation | 20 s | Field measurements |
| Precipitation, Temperature | Input data for hydrological simulation | 10 min | AEGIS-fire laboratory, University of the Aegean http://virtualfire.aegean.gr/ * |
| Digital Elevation Model (Dem) | HEC-GeoHMS terrain preprocessing | 5 m | Hellenic Cadastre http://gis.ktimanet.gr/ * |
| Land use | Parameters calculation for hydrological model | 1:10,000 | Northern Aegean Water Directorate http://www.apdaigaiou.gov.gr/ * |
| Soil | Parameters calculation for hydrological model | 1:1,000,000 | European Soil Data Centre (ESDAC) https://esdac.jrc.ec.europa.eu/ * |

**\* Accessed on 17 July 2023.**

Regarding the data-driven model, 11 different time series of hydro-meteorological features were investigated as inputs in the LSTM neural network models, and a comprehensive description of these input datasets can be found in Table 2. The target dataset used by the model was the recorded water level in the Kalloni telemetry station, sampled every 15 minutes since November 2018, aiming to forecast water levels for 15 min, 30 min,

45 min, and 1 h in advance within the study area. The dataset counting 140,257 records was divided into 50% training, 25% validation, and 25% test subsets. This partitioning strategy was tailored to account for the intermittent nature of the stream, which experiences both periodic and no-level periods across four full years. Therefore, to address the potentially conflicting effects associated with these distinct situations, two full years were allocated to the training set, while one year was dedicated to each of the validation and testing subsets. The training set was utilized for model fitting, the validation set was employed for adjusting hyperparameters and preventing overfitting [37], and the test set was used to assess the model's generalization ability. Following the dataset split, the input features underwent min–max normalization using Equation (2) to account for variations in magnitude across different features, thereby ensuring the preservation of information within the training set. By normalizing the data during training, more favorable outcomes can be achieved, such as improved results and reduced training time.

$$x_{normalization} = \frac{X - X_{min}}{X_{max} - X_{min}} \tag{2}$$

**Table 2.** Description of investigated inputs datasets for data-driven model.

| Feature | Description | Units |
| --- | --- | --- |
| Level | Target value: Level for each 15-min step | Meters (m) |
| MaxLevel48 | Maximum level of the previous 48 h | Meters (m) |
| Rain | Cumulative rainfall for each 15-min step | Millimeters (mm) |
| SumRain48 | Cumulative rainfall of the previous 48 h | Millimeters (mm) |
| Max48HrRain | Maximum hourly rainfall of the previous 48 h | Millimeters (mm) |
| SumRain7days | Cumulative rainfall of the previous 7 days | Millimeters (mm) |
| Intensity | Rain intensity | Millimeters/hour (mm/h) |
| Duration * | Rainfall duration up to the considered time | Hours (h) |
| DryPeriod | Dry period: cumulative hours of aridity | Hours (h) |
| Outflow | Discharge | Cubic meters/s ($m^3$/s) |
| Volume48 | Discharge volume of the previous 48 h | Cubic meters ($m^3$) |

* ASSUMPTIONS for Duration: 1. I consider it raining when the 15-min rainfall is >0.10 mm; 2. I consider the rain event to stop and the Duration to return to zero when there is no rain for the next 12 h; 3. When I have some intervening hours without rain (<12 h) the Duration keeps the same value until it rains again.

### 2.2.1. Trend and Seasonality of Level Target Dataset

This study utilizes time series decomposition to gain insights into water level fluctuations. Decomposition involves separating the series into trend, seasonality, and noise components, providing a comprehensive understanding of the underlying patterns. The trend component represents long-term developmental changes, while seasonality captures regular variations influenced by seasonal factors. The noise component accounts for incidental factors that introduce randomness into the data. By decomposing the water level time series, the model can better comprehend the complex dynamics of flood processes, considering the irregular and dynamic characteristics of individual data points [38].

As shown in Figure 3, the decomposition reveals three distinct components: long-term trends, seasonal fluctuations, and random residuals, each contributing to the overall behavior of the water level. The trend component exhibits consistent upward or downward slopes, with an annual peak observed between January and February. The seasonal component reveals a clear periodic pattern, indicating consistent seasonal changes. The residual component accounts for exceptional values or data gaps, reflecting the stochastic characteristics of the water level.

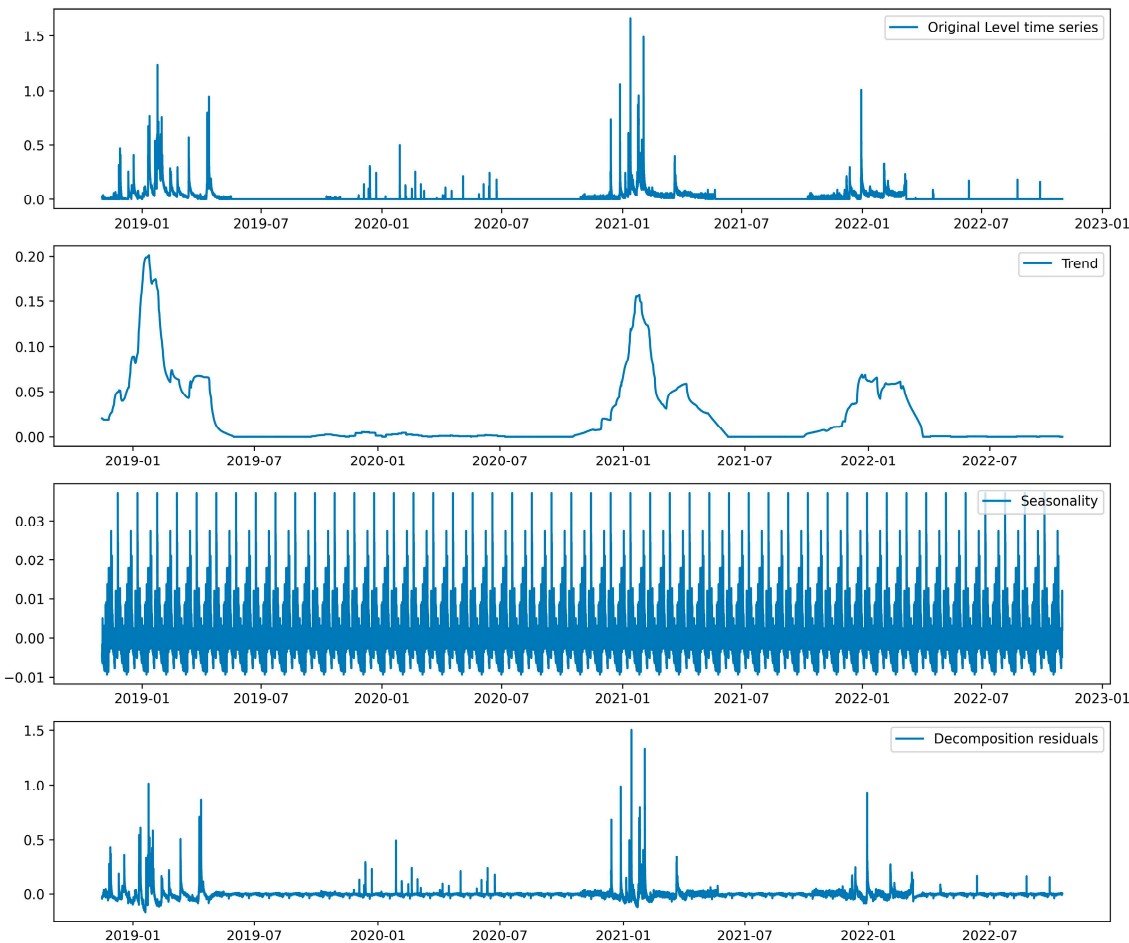

**Figure 3.** Original level and its decomposition in trend, seasonality, and residuals graphs.

### 2.2.2. Permutation Feature Importance

The selection of input variables in deep learning algorithms for forecasting poses a significant challenge, as it has a substantial impact on the model's performance. Proper feature selection is essential in reducing computational training time and improving model results [39]. To tackle this challenge, this study integrated feature importance techniques to identify the key parameters that have the most significant impact on predicting the target level. This procedure assigned scores to input features, as detailed in Table 2, based on their predictive value for the target variable. Various interpretation and analysis methods, such as statistical correlation, linear models, decision trees, and permutation importance, can be employed to determine the correlation between input–output variables.

In related studies, various strategies have been explored for feature selection in deep learning algorithms. Jamei et al. [40] utilized classification and regression trees (CART) to determine the most significant input variables. Jiang et al. [41] employed Expected Gradients for Feature Importance, while Liu et al. [42] utilized the gradient boosting regression tree (GBRT) to measure the importance of model inputs. GBRT, a recursive decision tree algorithm, constructs an ensemble of weak learners (decision trees) and combines their outcomes to provide the final prediction output.

Permutation feature importance (PFI) [43], utilized in this study, is a widely recognized method for identifying influential input variables. Previous studies [44–46] have employed PFI to interpret model behavior and assess the impact of ensemble members on predictions. By quantifying the improvement in model prediction error [47], PFI assigns random values to features based on their relationship with the model output [44]. Specifically, PFI quantifies the decrease in a model's score when the value of an individual feature is randomly shuffled [43]. The resulting decrease in model performance, measured by an

increase in model error, indicates the relative importance of the variable under consideration [46]. By randomly permuting the values of an input variable and evaluating the resulting degradation in model performance, PFI reveals the significance of the variable as far as the target variable is concerned [46]. This method enables the ranking of predictors, providing valuable information on the importance of each input feature in achieving accurate predictions [46,48]. PFI's estimation of importance enhances the interpretation of the influences of ensemble members on the LSTM model, assisting in understanding its behavior and serving as a valuable tool for identifying significant predictors prior to formulating a predictive model [45].

The present study adopted a customized variation of the permutation feature importance methodology, specifically tailored for LSTM models. This approach involved externally permuting the feature values prior to utilizing the LSTM model for prediction, as opposed to permuting within the model's architecture. Despite variations in implementation, the fundamental principle remained unaltered: assessing feature importance by quantifying the resulting changes in model performance. To compute feature importance, the LSTM model's performance on the original data was compared with its performance after permuting the values of each feature, utilizing mean squared error as the scoring metric. Iterating through each feature, the values were permuted, and the LSTM model generated predictions using the permuted data. These predictions were subsequently compared against the true target values, and the reduction in performance relative to the baseline was calculated to determine the feature importance scores. These scores were stored for subsequent analysis, facilitating a comprehensive evaluation of each feature's significance within the LSTM model. Hence, this approach can be acknowledged as a variant or adaptation of the permutation feature importance methodology specifically designed for LSTM models. Furthermore, two rounds of attribute importance analysis were conducted, one including the target parameter "Level" and another omitting it due to its significant impact on prediction outcomes. To interpret the models, multiple tools, including permutation importance, were utilized through the Python Scikit-Learn library, enabling the ranking and interpretation of the attributes outlined in Table 2 and thereby enhancing the comprehensibility of the obtained results.

### 2.3. Physical Model: HEC-HMS

The Hydrologic Engineering Center's Hydrologic Modeling System (HEC-HMS) is a physically-based, semi-distributed, deterministic, conceptual, hydrological model that simulates runoff generation processes of dendritic drainage basins by taking into account sub-catchments [3]. Developed by the US Army Corps of Engineers (USACE), HEC-HMS is widely used in the hydrological research area due to its straightforwardness, free accessibility, and ability to integrate spatial data [49]. It is particularly suitable for flood forecasting in situations with limited data inputs, owing to its easy calibration process [32]. Additionally, HEC-HMS is a simple yet accurate modeling approach that can predict the temporal and spatial responses of watersheds to various events, including short- and long-term ones, under diverse climate and soil conditions [50,51]. It has been shown to be effective in modeling hydrologic processes across a diverse range of geographic regions and scales, from small urban basins [22,49] to large-scale basins [52,53]. Studies have demonstrated the model's ability in conducting continuous [54] and event-based simulations [55] with reliable results.

Combining GIS tools with hydrological modeling has improved the accuracy of hydrological models, particularly by providing new platforms for data management and visualization, and by streamlining data input and enhancing the interpretation of model outputs [49]. GIS applications are utilized to generate hydrological data such as flow direction, flow accumulation, and basin and stream network delineation, from a digital elevation model [22]. In this study, the HEC-HMS 4.9 model was integrated with GIS using the HEC-GeoHMS 10.4 extension module in ArcGIS 10.4 software. The cartographic materials were used as input data to generate a numerical database of the Kalloni river

catchment. The HEC-GeoHMS toolbox was used to preprocess the DEM as well as land use and soil maps, enabling a calculation and delineation of the physical and drainage characteristics of the Kalloni watershed. In addition, some required parameters for HEC-HMS simulation, such as maximum storage, constant loss rate, and imperviousness rate in each sub-basin, were derived from land use polygons and hydrological soil groups categorized in HEC-GeoHMS. The catchment area was partitioned into semi-distributed sub-basins, and all the required model parameters were calculated and input into HEC-HMS for the initial simulation.

The hydrologic elements of the Kalloni basin model consist of 21 sub-basins, 12 junctions, and 11 main reaches. Meteorological and field data were imported through the HEC-DSSVue database. Sub-hourly 15-min data including precipitation, observed streamflow, and evapotranspiration data were input in the meteorological model for the continuous four-year simulation from 2018 to 2022, as specified in the HEC-HMS model control specifications. In addition, a warm-up period was necessary to achieve the dynamic equilibrium of hydrological models [56,57]. To ensure proper warming up of the model in this study, a consecutive one-year cycle including the years 2017–2018 was incorporated into the simulation process. To maintain the model' calibration and validation process, gauge discharge information from the telemetry station in Kalloni settlement, incorporated into real-time series data, was utilized at Junction 110. Calibration was conducted from 1 November 2018 to 1 November 2019, and validation was performed between 1 November 2020 and 1 November 2021, with a temporal resolution of 15 min.

HEC-HMS divides the hydrological cycle into individual parts, enabling each component such as surface runoff, infiltration, evaporation, transpiration, and precipitation to be represented by a separate mathematical model [3,58]. Regarding each basin, in order to calculate the excess rainfall, HEC-HMS uses a set of equations that represent the rainfall loss, or a transformation model that converts the excess rainfall into direct runoff, which assumes steady and uniform geographical distribution [53,59]. HEC-HMS includes several traditional hydrological analysis methods essential for soil moisture accounting, evapotranspiration, and snowmelt [60]. The model also includes 12 distinct loss estimation methods, seven methods for transforming rainfall-runoff, five baseflow methods, and eight methods for routing that are specifically developed to perform simulations of individual events, or tailored for continuous simulation purposes [21,53]. In this study, hydrologic methods that have been proven effective in continuous simulation and require minimal parameters were selected based on their simplicity and ability to account for watershed storage, timing, and capability to simulate long-term periods.

The Simple Canopy approach was chosen to represent vegetation in the landscape due to its widespread use in plant canopy representation [57]. To demonstrate surface runoff occurrence when rainfall surpasses the rate of infiltration and the storage capacity at the surface is reached, the Simple Surface method was selected [57]. Existing literature indicates that surface storage values are influenced by basin slope and land use types [61]. Furthermore, the deficit and constant method was used to compute infiltration losses, with parameters such as maximum storage, constant rate, initial deficit, and imperviousness derived from available cartographic data. The Clark Unit hydrograph method, which requires only two easily assessable parameters (i.e., time of concentration ($t_C$) and storage coefficient (R)), was chosen among available transform methods to transform precipitation excess into direct surface runoff at the basin outlet. The storage coefficient (R) represents the time excess precipitation is stored within the watershed as it flows towards the outlet location [55]. This coefficient was initially calculated as a percentage of the concentration time, with the final estimate obtained through calibration. The $t_C$ parameter was determined by using the HEC-HMS Handbook formula:

$$t_c = \frac{l^{0.8} \times (S+1)^{0.7}}{1140 \times Y^{0.5}} \quad (3)$$

where, l (ft) is the length of the hydraulically longest flow path; Y (%) is the watercourse slope of the longest flow path; and S (in) is the potential maximum retention, which reads as

$$s = \frac{1000}{CN} - 10 \tag{4}$$

In this approach, the curve number (CN) was assigned to sub-basins based on the hydrologic soil group and land use type, with the estimation carried out manually. By considering the relevant soil and land cover characteristics, the CN value was determined for each unit within the sub-basin and then aggregated using area-weighting. The tables provided in the Technical Release Number 55 were utilized for the computation process [62].

In addition, the model utilized the Linear Reservoir method with two groundwater layers to simulate baseflow, which is the sustained runoff of previously stored precipitation that temporarily resides in the basin before flowing into the channel [55]. This method accounts for an underground storage reservoir that accumulates rainwater during the infiltration phase and subsequently discharges, thus contributing to the surface runoff after the rainfall cessation [3]. The Lag procedure was adopted for flow routing, with the outflow hydrograph being comparable to the inflow but delayed in time, and a velocity of 3 m/s was assumed for the lag time of every sub-catchment. Finally, incorporating the evapotranspiration process in the HEC-HMS model is crucial for long-term simulations and when employing the deficit and constant loss approach [3]. The constant monthly evapotranspiration method was chosen in this study, which requires potential monthly evaporation rates and crop coefficients for all sub-basins. The mean daily potential evapotranspiration was computed using the modified Blaney–Criddle approach, as explained by Koutsovili et al. [3]. Table 3 provides the calculation methods utilized in the present study for all components of the HEC-HMS model, as well as the required input parameters.

**Table 3.** Selected methods and input parameters for HEC-HMS model components.

| Component | Method | Parameter | Unit |
|---|---|---|---|
| Canopy | Simple Canopy | Initial Storage<br>Max Storage<br>Crop Coefficient | %<br>mm<br>- |
| Surface | Simple Surface | Initial Storage<br>Max Storage | %<br>mm |
| Loss | Deficit and Constant | Initial Deficit<br>Maximum Deficit<br>Constant Rate<br>Impervious | mm<br>mm<br>mm/h<br>% |
| Transform | Clark Unit Hydrograph | Time of concentration<br>Storage Coefficient | h<br>h |
| Baseflow | Linear Reservoir | GW 1 Initial<br>GW 1 Fraction<br>GW 1 Coefficient<br>GW 2 Initial<br>GW 2 Fraction<br>GW 2 Coefficient | $m^3/s$<br>-<br>h<br>$m^3/s$<br>-<br>h |
| Routing | Lag | Lag Time | min |
| Evapotranspiration | Constant Monthly | Monthly Evaporation Rate<br>Crop Coefficient | mm/month<br>- |

Calibrating a hydrological model with the relevant data constitutes an important step as far as the accurate representation of a basin is concerned [52]. The calibration process involves modifying parameters such as infiltration and storage coefficient, and baseflow parameters to achieve the best fit between simulated and observed results. The input data's quality and the technical abilities of the hydrological model may also affect the

effective model's calibration [58]. Herein, the HEC-HMS method underwent a manual calibration and verification process using previously observed hydrological data to predict streamflow at a 15-min interval. The calibration process involved adjusting parameter values iteratively until the model's results matched the observed data, ensuring parameters remained within a reasonable range [63]. Calibration aimed to achieve consistency between the computed and observed discharge data in terms of curve shape, and value and time of peak. Subsequently, the same fine-tuned parameters obtained during calibration were used for model validation. Validation involved generating flood hydrographs for the catchment area and calculating goodness-of-fit indices to assess the concurrence between modeled and observed hydrographs. By leveraging the Calibration Aids tool within the HEC-HMS model, several parameters of the model were determined through empirical or manual estimation. The accuracy and reliability of the model were evaluated in terms of statistical analysis to assess certain performance criteria used to predict the peak flows, the total hydrograph volume, and the time to peak [64]. Both the calibration and validation stages relied on a long period of observed flow to ensure the model's consistent performance in continuous runoff simulation. This comprehensive process of calibration and validation contributed to improving the predictability and reliability of the HEC-HMS model in streamflow prediction tasks.

### 2.4. Data-Driven Model: Long Short-Term Memory

Machine learning has gained significant popularity in forecasting floods, offering a robust data-driven approach without the need for explicit knowledge of complex nonlinear dynamic processes [65,66]. Artificial neural networks (ANNs) have been widely adopted in hydrology and other domains due to their effectiveness and learning capabilities. Neural networks comprise interconnected nodes or neurons arranged in several layers [39]. Among these, recurrent neural networks (RNNs) have emerged as a popular model designed to leverage time series data and handle long input sequences [67,68]. Structurally, RNNs consist of an input layer, an output layer, and one or more hidden layers, with a distinct feedback recurrent layer facilitating the retention of information across multiple steps. RNNs are often referred to as "backpropagation through time" due to their utilization of the backpropagation algorithm for gradient calculation, weight matrix adjustment, and weight updates during the feedback process [14]. In time series forecasting, the historical data's patterns play a crucial role in accurate prediction [69]. RNNs utilize feedback connections to retain information from past inputs, capturing temporal dynamics, while their hidden state enables them to capture dependencies among sequential data elements, maintaining the relationship between past and current observations [67,70]. However, RNNs face challenges related to vanishing gradients when learning long-range dependencies, limiting their long-term forecasting capability [71]. To address this limitation, Long Short-Term Memory (LSTM) was introduced to incorporate memory cells to regulate information flow improving the performance of typical RNNs [72–74].

First introduced by Hochreiter and Schmidhuber [73], LSTM stands as an advanced variation of the RNN architecture, because it possesses a remarkable capability to capture both long-term and short-term dependencies, with its memory cell playing a crucial role in storing and retaining cell states [74]. LSTM networks employ gates to enhance their performance [75]. These gates, including input, output, and forget gates, play a crucial role in remembering and learning from past information, thereby facilitating accurate time series prediction of sequential data [72,76]. The forget mechanism selectively discards specific historical information, while retaining and integrating new updates with historical information during the backward transfer [77]. In the field of hydrological time series analysis, LSTM has emerged as a powerful tool, leveraging the key components of RNN memory cells, such as input, self-recurrent connection, forget, and output gates, to establish a robust framework [65,78]. Additionally, LSTM networks have been widely acknowledged for their superior performance in multi-step predictions of time series data, as evidenced by studies conducted by Kratzert et al. [79] and Yunpeng et al. [31].

This research specifically focuses on investigating the impact of different input sequences on various LSTM architectures and proposes a sensitive LSTM architecture along with an optimized input sequence. It employs a multiple output mechanism for LSTM, enabling them to use one-shot procedures to predict a sequence of values. Hydrological time series are involved in training and validating LSTM for short-term leading times ranging from 15 min to 1 h.

### 2.4.1. Structure of LSTM Architectures

Long Short-Term Memory (LSTM) models, which have gained widespread recognition as effective forecasting models for hydrological time series [65], encompass a range of developed and widely utilized architectures in time series forecasting tasks. In this study, five different LSTM architectures were specifically investigated: (i) Vanilla LSTM, (ii) Stacked LSTM, (iii) Bidirectional LSTM, (iv) Encoder–Decoder LSTM, and (v) Encoder–Decoder Bidirectional LSTM.

The Vanilla LSTM architecture is a widely used and straightforward design for time series forecasting, featuring a single LSTM layer succeeded by one or multiple fully connected Dense layers. Within the LSTM layer, distinct gates, namely the forget gate, input gate, and output gate, regulate the information flow. The input gate regulates the extent to which new state information is utilized, while the output gate controls the quantity of information extracted from preceding states [67]. Additionally, the forget gate governs the retention or elimination of internal state information for propagation to subsequent layers. Furthermore, LSTM employs an internal connection mechanism through a multiplication gate, allowing the model to learn and determine when to reset the memory contents using another unit [76]. Compared to traditional recurrent neural networks, LSTM incorporates an additional cell state, denoted as C, facilitating the retention of long-term information. The cell state is dynamically updated at each time step through the coordinated operation of the forget gate and the input gate, allowing LSTM to effectively retain information over extended periods without suffering from the issue of vanishing gradients [80]. Figure 4 illustrates the Vanilla LSTM architecture, depicting the long-term memory ($C_t$) and short-term (hidden) memory ($h_t$) within the cell. LSTM assigns weights and biases to input hidden layer values, employs activation functions to determine node outputs, and optimizes weight matrices and bias vectors to minimize a pre-defined error during the training process [39]. The fundamental equations below provide the basic definitions for LSTM neural networks.

$$f_t = \sigma \left( W_f X_t + U_f h_{t-1} + b_f \right) \tag{5}$$

$$I_t = \sigma \left( W_i X_t + U_i h_{t-1} + b_i \right) \tag{6}$$

$$O_t = \sigma \left( W_o X_t + U_o h_{t-1} + b_o \right) \tag{7}$$

$$\widetilde{C}_t = \tanh \left( W_c X_t + U_c h_{t-1} + b_c \right) \tag{8}$$

$$C_t = f_t \times C_{t-1} + I_t \times \widetilde{C}_t \tag{9}$$

$$h_t = O_t \times \tanh(C_t) \tag{10}$$

$$y_t = V C_t + b_y \tag{11}$$

where, $f_t$, $I_t$, and $O_t$ are activation vectors for the forget, input, and output gates, respectively, at time t; $X_t$ is the input at time step t; $W_i$ and $U_i$ represent weight matrices; $b_i$ corresponds to bias vectors feeding into the hidden and output layers; σ denotes the chosen activation function; $\widetilde{C}_t$ denotes the candidate for the cell-state value; $C_t$ and $h_t$ indicate the current cell and the hidden state, respectively; $y_t$ is the output of the time step t, and V represents the weight matrix connecting the hidden layer and the output layer.

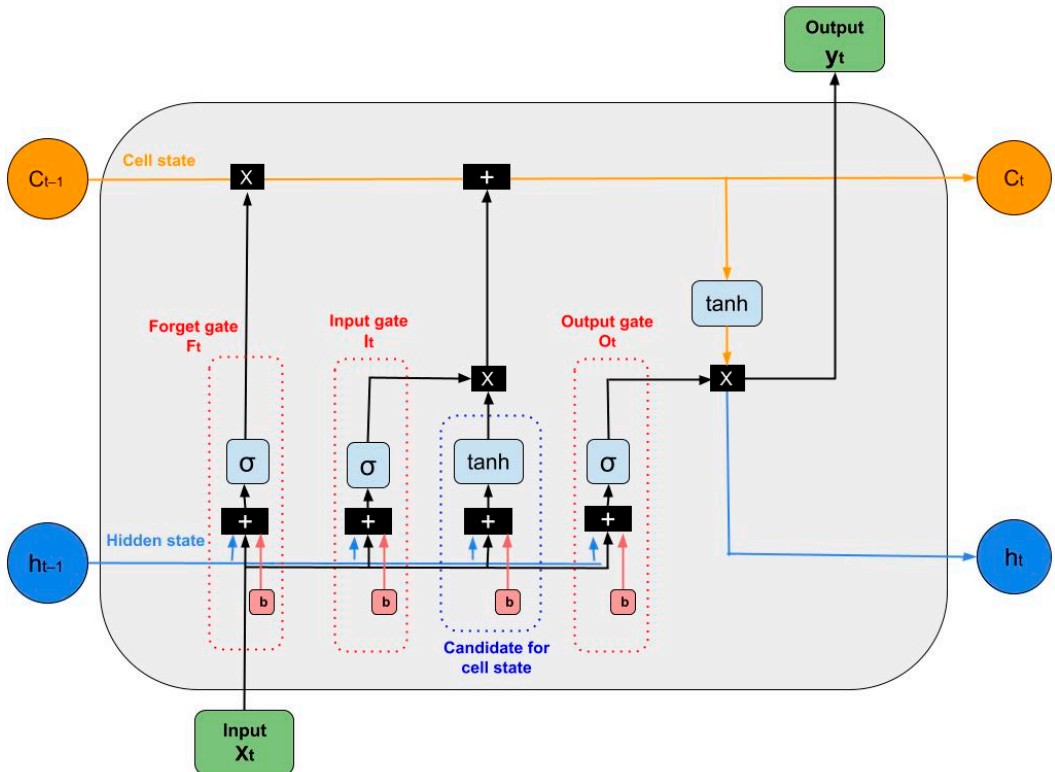

**Figure 4.** The structure of the Vanilla LSTM model.

This research explored additional LSTM architectures in order to analyze their complexity and performance. The Stacked LSTM architecture involves the incorporation of multiple LSTM layers that are sequentially stacked on top of each other. Each LSTM layer takes the output sequence of the previous layer as input, allowing for the capture of more intricate temporal dependencies within the data. By progressively adding hidden layers, it generates abstract representations of the sequence data, enhancing effectiveness and reducing training time [81]. Moreover, Stacked LSTM networks, with multiple LSTM networks connected successively, provide higher representation for time series data compared to individual LSTM networks in certain applications [67,82].

Bidirectional LSTM is an advanced architecture that analyzes input sequences in both forward and backward directions, effectively capturing dependencies from both past and future time steps and improving future value forecasting. By incorporating information from both directions, Bidirectional LSTM networks gain a deeper understanding of the contextual information surrounding predictions [67]. Unlike standard RNN cells that sequentially process data from left to right, Bidirectional LSTM employs two sets of hidden layers: forward states and backward states [12]. These two hidden layers operate independently but receive the input value, enabling comprehensive calculations. The output value is determined by considering the data from both hidden layers. The forward calculation follows the standard RNN approach, while the input values of the backward hidden layer are processed in the opposite direction, resulting in a comprehensive output layer calculation [12]. Bidirectional LSTM networks traverse input sequences twice, once in the past-to-future direction and then in the future-to-past direction, allowing for enhanced training and capturing of bidirectional dependencies [68].

While LSTM networks capture long-term dependencies, they typically need an equal number of time steps for elaborating on input–output data [68]. To address this limitation, researchers have developed sequence-to-sequence (Seq2Seq) learning mechanisms, which enable the transformation of sequences from one domain to another in order to address sequence-based tasks [83]. In an LSTM Seq2Seq model, multiple gates are employed to facilitate memory retention of past data. For sequence-to-sequence prediction tasks with

input and output sequences of varying lengths, the commonly used Encoder–Decoder architecture is adopted. In this study, two additional architectures are investigated: a simple Encoder–Decoder LSTM and an Encoder–Decoder Bidirectional LSTM. The Encoder–Decoder LSTM architecture comprises two components: an encoder LSTM that processes the input sequence into a fixed-length vector and a decoder LSTM that generates the output sequence. A time-distributed layer follows the decoder layer and applies a dense transformation to each time step in the output sequence. Overall, this architecture takes a sequence of inputs, encodes it using an LSTM layer or a Bidirectional LSTM layer, and decodes it to produce a sequence of outputs. This model's innovative aspect lies in its utilization of a fixed-sized internal representation, facilitating efficient reading of input sequences and generating corresponding output sequences [84]. In addition, the Encoder–Decoder scheme has proven effective in reliable and accurate multi-step-ahead flood forecasting. By utilizing sequence-to-sequence learning, it employs LSTM units to build a deep learning neural network with a multi-input and multi-output model structure [83]. This integrated approach enhances sequence prediction performance and offers valuable insights for diverse applications.

### 2.4.2. Implementation and Settings of LSTM Models

In this paper LSTM models were implemented on the Keras library with TensorFlow backend [85]. Keras is a Python interface developed by Google that offers an open-source software for artificial neural networks and deep learning [75,86]. Additionally, the implementation utilized the Pandas, NumPy, Scikit Learn, and Matplotlib libraries to address specific data-driven modeling requirements.

The implementation of LSTM models begins by defining the supervised learning problem, involving the selection of input and output time steps, and the predictor and target features. Based on experimental investigations, 4-h input sequences were identified as optimal for forecasting at different time intervals. Subsequent to an iterative testing process, it was discerned that the utilization of multiple lag time steps enhances the long-term predictive performance. Nevertheless, prudent consideration is warranted when approaching the upper limit of input time steps, taking into account factors related to computational efficiency and the potential for overfitting. A permutation feature importance technique using the Scikit-Learn package was applied to rank the predictor features based on their impact. The dataset was then divided into 50% training, 25% validation, and 25% testing subsets. To ensure consistent scaling, the input features and target datasets were normalized using the MinMaxScaler function. This normalization process, ranging from 0 to 1, addresses the variability in feature values. The datasets were structured and prepared for training the level prediction models, with the final outputs rescaled and inverted to obtain the original values. The subsequent phase involved defining and fitting the LSTM models with specific configurations. A multivariate input setup was adopted, with an input layer of 16 time steps representing a 4-h lag time and selected feature predictors. The output layer enclosed four neurons with linear activation for predicting multi-step outputs. The models were trained for 80 epochs using an Adam optimizer, Mean Squared Error (MSE) loss function, and Root Mean Squared Error (RMSE) as the evaluation metrics. A learning rate scheduler was implemented to adjust the learning rate throughout training.

In order to optimize the performance of the LSTM models, a thorough hyperparameter tuning process was conducted during the training phase. This process leveraged the GridSearchCV function from the sklearn module, enabling systematic exploration of various combinations of hyperparameters. Notably, the GridSearchCV constructor incorporates a cross-validation parameter, which determines the number of folds or subsets created during cross-validation. In this study, the dataset was divided into three equal-sized folds, resulting in the model being trained and evaluated three times. Cross-validation ensured a more reliable estimation of the model's performance by mitigating the impact of data variability and bias that may arise from a specific train–test split. By employing cross-

validation, the model's performance became less dependent on any particular subset of the data.

To optimize the performance of the LSTM models, multiple hyperparameter aspects were investigated. These aspects included the number of neurons in the hidden layer, batch size, activation function, and learning rate hyperparameters. The range of neurons tested spanned from 40 to 400, with larger numbers generally associated with better performance but increased computational cost. Batch sizes varied from 16 to 63, considering the trade-off between training speed and model quality. A too-large batch size may compromise model quality, while too-small batch size may hinder convergence [68]. Various activation functions, namely hyperbolic tangent (tanh), rectified linear unit (ReLU), and logistic sigmoid, were examined for the LSTM neurons. Additionally, various learning rates (0.1, 0.01, 0.001, and 0.0001) were investigated to determine the optimal value. The findings consistently demonstrated the selection of the hyperbolic tangent function (tanh) as the activation function, and a learning rate of 0.001 yielded superior performance across all models. For the hyperparameters that were not tuned, default values based on previous studies and established heuristics in the field of Deep Learning were employed, ensuring informed choices in those cases. Table 4 provides a detailed summary of the selected hyperparameters and the breakdown of the five LSTM architectures investigated in this research.

**Table 4.** Breakdown of the examined LSTM architecture and hyperparameters.

| Name | Vanilla LSTM | Stacked LSTM | Bidirectional LSTM | Encoder–Decoder LSTM | Encoder–Decoder Bidirectional LSTM |
|---|---|---|---|---|---|
| Model | Sequential | Sequential | Sequential | Encoder-Decoder | Encoder-Decoder |
| LSTM hidden layers | 1 | 2 | 1 | 1 Encoder 1 Decoder | 1 Encoder 1 Decoder |
| LSTM units/memory cells | 48 | 1st 48 2nd 64 | 96 | Encoder 40 Decoder 40 | Encoder 200 Decoder 400 |
| LSTM activation function | tanh | tanh | tanh | tanh | tanh |
| Dense layers | 1 | 1 | 1 | 1 | 1 |
| Dense units/memory cells | 4 | 4 | 4 | 4 | 4 |
| Dense activation function | Linear | Linear | Linear | Linear | Linear |
| Optimizer | Adam | Adam | Adam | Adam | Adam |
| Learning rate | 0.001 | 0.001 | 0.001 | 0.001 | 0.001 |
| Loss function | MSE | MSE | MSE | MSE | MSE |
| Evaluation metric | RMSE | RMSE | RMSE | RMSE | RMSE |
| Batch size | 32 | 28 | 40 | 16 | 28 |
| Epochs | 80 | 80 | 80 | 80 | 80 |

### 2.5. Model Evaluation Criteria

In the final stage of the methodology, the performances of both the physical HEC-HMS model and the data-driven model were evaluated using a set of various statistical indicators. For the physical HEC-HMS model, the calibration and validation accuracy and performance were assessed using four commonly used goodness-of-fit measures. These measures included the Nash–Sutcliffe efficiency coefficient (NSE), the percentage bias error (PBIAS), the Root Mean Squared Error standard deviation ratio (RMSE Std. Dev.), and the Coefficient of Determination ($R^2$). The utilization of these parameters is a prevalent practice in hydrology to evaluate the correlation between predicted and observed results [65]. These statistical indicators were employed to gauge the HEC-HMS model's statistical performance at a 15-min interval. A perfect match between the simulated and observed values would yield NSE, PBIAS, RMSE Std. Dev., and $R^2$ values of 1, 0%, 0, and 1, respectively. The calculation of these evaluation metrics was based on the following Equations (12)–(15).

Nash–Sutcliffe efficiency coefficient (NSE), given by [87]:

$$\text{NSE} = 1 - \frac{\sum_{i=1}^{n} \left( Q_i^{\text{sim}} - Q_i^{\text{obs}} \right)^2}{\sum_{i=1}^{n} \left( Q_i^{\text{obs}} - \overline{Q^{\text{obs}}} \right)^2} \tag{12}$$

Percentage bias error (PBIAS), given by:

$$\text{PBIAS} = \frac{\left| \overline{Q^{\text{sim}}} - \overline{Q^{\text{obs}}} \right|}{\overline{Q^{\text{sim}}}} \times 100 \ (\%) \tag{13}$$

Root Mean Squared Error standard deviation (RMSE Std. Dev.) reads as

$$\text{RMSE Std. Dev.} = \frac{\sqrt{\sum_{i=1}^{n} \left( Q_i^{\text{obs}} - Q_i^{\text{sim}} \right)^2}}{\sqrt{\sum_{i=1}^{n} \left( Q_i^{\text{obs}} - \overline{Q^{\text{obs}}} \right)^2}} \tag{14}$$

Coefficient of determination ($R^2$) is as follows:

$$R^2 = \left[ \frac{\sum_{i=1}^{n} \left( Q_i^{\text{obs}} - \overline{Q^{\text{obs}}} \right) \left( Q_i^{\text{sim}} - \overline{Q^{\text{sim}}} \right)}{\sqrt{\sum_{i=1}^{n} \left( Q_i^{\text{obs}} - \overline{Q^{\text{obs}}} \right)^2} \sqrt{\sum_{i=1}^{n} \left( Q_i^{\text{sim}} - \overline{Q^{\text{sim}}} \right)^2}} \right]^2 \tag{15}$$

where: $Q_i^{obs}$ and $Q_i^{\text{sim}}$ are the observed and simulated discharge value at the ith step, respectively; $\overline{Q^{obs}}$ and $\overline{Q^{\text{sim}}}$ are the observed and simulated mean discharge, respectively; and n is the number of observed/simulated values.

Furthermore, to assess the accuracy and predictive performance of the LSTM models, four performance indicators were employed. These indicators comprised the Root Mean Squared Error (RMSE), Root Mean Squared Logarithmic Error (RMSLE), coefficient of determination ($R^2$), and mean absolute error (MAE). These metrics are widely recognized and utilized for evaluating the quality of time series forecasting outcomes. The RMSE, RMSLE, and MAE were calculated using Equations (16)–(18).

Root Mean Squared Error (RMSE), given by:

$$\text{RMSE} = \sqrt{\frac{1}{n} \sum_{i=1}^{n} \left( L_i^{\text{pred}} - L_i^{\text{obs}} \right)^2} \tag{16}$$

Root Mean Squared Logarithmic Error (RMSLE), given by:

$$\text{RMSLE} = \sqrt{\frac{\sum_{i=1}^{n} \left( \log\left( L_i^{\text{pred}} + 1 \right) - \log\left( L_i^{\text{obs}} + 1 \right) \right)^2}{n}} \tag{17}$$

Mean absolute error (MAE), given by:

$$\text{MAE} = \frac{\sum_{i=1}^{n} \left( \left| L_i^{\text{pred}} - L_i^{\text{obs}} \right| \right)}{n} \tag{18}$$

where:
$L_i^{\text{pred}}$ and $L_i^{obs}$ are the predicted and observed level value at the ith step, respectively; and n is the number of predicted/observed values.

## 3. Results

### 3.1. Hydrological Analysis Based on the Physical Model HEC-HMS

During the physical HEC-HMS simulation process, the pre-set parameter values were fine-tuned through calibration by comparing the simulated results with observed data. Subsequently, the model was validated using the calibrated parameter values, and the obtained results were evaluated statistically. This process aimed to verify that the simulated discharge values matched the historical telemetry data at gauge stations in the basin, while also considering acceptable range indicators and the statistical performance of the model. The calibration procedure demonstrated its effectiveness through visual and statistical comparisons.

Figures 5 and 6 compare the observed and the predicted discharges during the calibration (November 2018–November 2019) and validation (November 2020–November 2021) periods using the HEC-HMS physical model. Figure 5 specifically illustrates the flow hydrograph of simulated and observed discharges throughout the two periods. The graphical comparison revealed a remarkable resemblance between the observed and the predicted runoff hydrograph. The results demonstrated a similarity between the original and predicted discharges on a 15-min time scale, indicating the model's effectiveness in capturing the dynamics of the system.

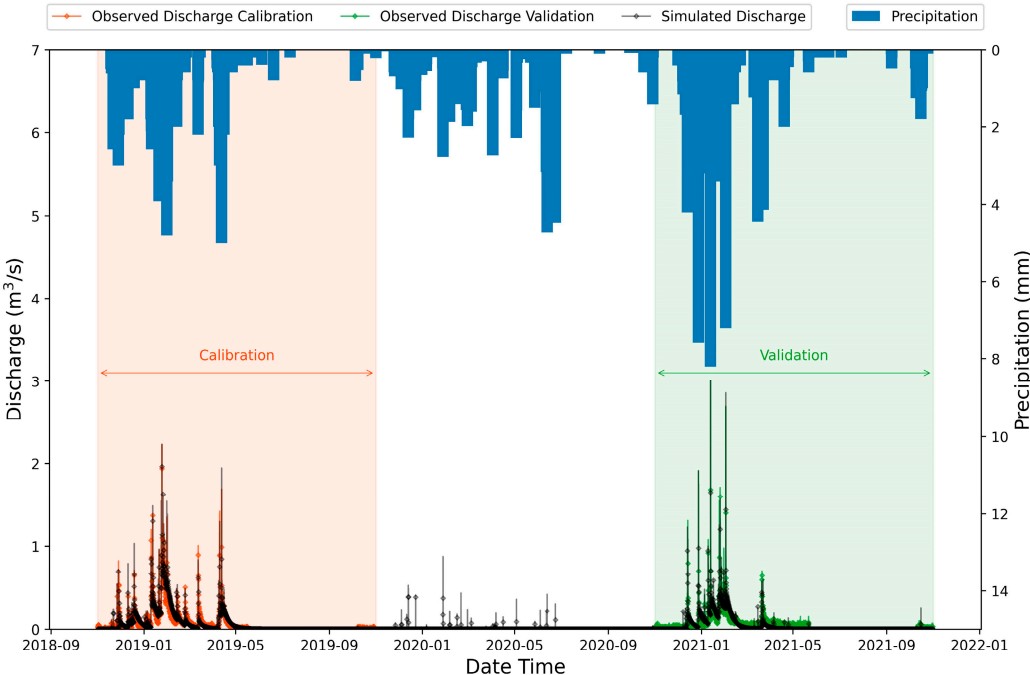

**Figure 5.** Observed and simulated hydrographs for the calibration and validation periods using the HEC-HMS physical model.

Furthermore, Figure 6 presents scatter plots depicting the correlation between the observed and simulated discharge results for both the calibration and validation period. The scatter plots include colored bars indicating the residuals. The distribution of simulated and observed values exhibited a satisfactory dispersion along both the uphill and downhill directions, and the trend line almost approximated the 1:1 line. A majority of the points aligned closely with the 1:1 line, indicating a high degree of agreement in the model's predictive capabilities during both periods. Remarkably, the scatter plot for the validation period displayed smaller deviations that closely approximated the 1:1 line compared to the calibration period. The graphs revealed a uniform distribution of the simulated and observed discharge values around the trend line, providing evidence of a strong correlation between the two datasets and underscoring the effectiveness of the model.

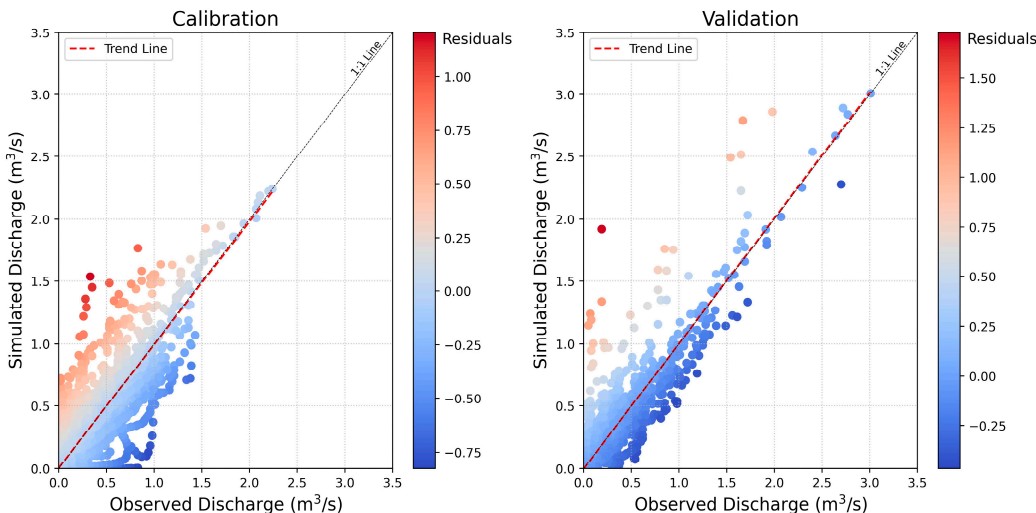

**Figure 6.** Scatter plots with colored bars for residuals, comparing simulated and observed discharge for the calibration and validation periods.

In addition to visual comparisons, it is essential to evaluate the numerical outcomes and assess their statistical performance. Therefore, an analysis of peak streamflow, peak time, and flood volume was conducted to evaluate the calibration and validation outcomes. Table 5 provides a comprehensive overview of the simulated results, observed values, and corresponding residuals for peak discharge, date of peak discharge, and volume, both during the calibration and validation runs of the HEC-HMS model. Notably, the model demonstrated an exact correspondence between the simulated and observed values for the date, time, and magnitude of peak discharge in both the calibration and validation phases. The residuals corresponding to the streamflow volume displayed minimal variations between the simulated and observed results.

**Table 5.** HEC-HMS simulation results during calibration and validation.

| Variable | Calibration | | | Validation | | |
|---|---|---|---|---|---|---|
| | Observed | Simulated | Residual | Observed | Simulated | Residual |
| Peak Discharge ($m^3/s$) | 2.24 | 2.24 | 0.000 | 3.01 | 3.01 | 0.001 |
| Volume ($10^3$ $m^3$) | 2029.437 | 2174.594 | 145.157 | 1576.701 | 1463.063 | 113.638 |
| Date of peak | 24 January 2019, 08:15 | 24 January 2019, 08:15 | - | 12 January 2021, 21:30 | 12 January 2021, 21:30 | - |

Moreover, four statistical indicators were computed to further evaluate the performance of the HEC-HMS model. The Nash–Sutcliffe efficiency (NSE) coefficient was employed to measure the agreement between the model and observed data, with values exceeding 0.65 indicating satisfactory calibration and values surpassing 0.75 indicating very good calibration [88]. The percentage bias error (PBIAS) was utilized to assess the model's ability to maintain water balance, and a PBIAS value lower than ±10% was considered indicative of a very good calibration [88]. The Root Mean Squared Error standard deviation ratio (RMSE Std. Dev.) incorporated error index statistics and normalization, and values below 0.50 were indicative of very good calibration [88]. The coefficient of determination ($R^2$) provided a measure of the variance between the observed and simulated data, with values closer to 1 indicating a better fit of the model, and values greater than 0.7 being generally considered acceptable.

Table 6 presents the calculated values and the corresponding performance evaluation based on the assessed statistical metrics for both the calibration and validation phases. The analysis of the Percentage Bias Error (PBIAS) revealed that the model overestimated the runoff volume by 6.68% during the calibration phase and underestimated it by 7.77%

during the validation phase. These values indicated a very good performance of the model in accurately predicting the runoff volume. Overall, the statistical results indicated a range of performances from very good to good in both phases, underscoring the successful calibration and validation of the HEC-HMS model. In summary, these findings highlighted a high level of agreement between the original and predicted outcomes, providing strong evidence for the model's reliable performance.

**Table 6.** Statistical performance metrics during calibration and validation in HEC-HMS.

| Variable | Calibration | | Validation | |
|---|---|---|---|---|
| | Value | Performance | Value | Performance |
| NSE | 0.77 | Very good | 0.74 | Good |
| PBIAS | 6.68% | Very good | −7.77% | Very good |
| RMSE Std DEV | 0.48 | Very good | 0.51 | Good |
| R2 | 0.81 | Very good | 0.80 | Very good |

*3.2. Feature Importance Investigation*

The feature importance investigation in this methodology framework consisted of three stages. In the first stage, the models were configured to perform univariate forecasting, where the input consisted solely of lagged time step values of the Level variable, which was also the target variable. This approach allowed for an initial assessment of the model's predictive capabilities. In the second stage, the entire set of 11 investigated features was introduced as inputs to the model, aiming to examine the potential improvement in predictive performance with the inclusion of additional variables. This analysis provided valuable insights into the contribution of each feature to the forecasting task. Finally, in the third stage, the models were fitted with the target Level features along with the four top-ranked features identified through the permutation feature importance analysis. This stage focused on refining the model inputs by incorporating the most influential variables, further enhancing the accuracy and dependability of the forecasting results.

This study employed a modified version of the Permutation Feature Importance (PFI) method specifically designed for LSTM models to rank the input variables and gain insights into their impact on the forecasting task. The utilization of tools such as permutation importance, implemented through the Python Scikit-Learn package, enhanced the interpretability of the results. Figure 7 illustrates the importance of 10 ensemble input features of the LSTM model during the calibration period, providing valuable insights into the model's behavior. The feature importance values were presented in the form of a bar graph, visually representing the individual impact of each feature on the prediction of the target variable. The feature importance values indicated the extent to which an increase in a specific feature value positively or negatively affects the prediction of the target variable. A positive feature importance value signifies that an increase in the corresponding feature value positively contributes to the prediction, while a negative value suggests the opposite.

The analysis of the results revealed that certain features exhibited varying levels of importance in the model's performance. Among the evaluated features, the Outflow feature demonstrated the highest positive feature importance value, indicating its significant impact on the model's predictions (importance score: 0.303). Following closely was the MaxLevel48 feature, which captured critical information about maximum water levels within a 48-h period and exhibited substantial importance (importance score: 0.252). The DryPeriod and Max48HrRain features also held notable importance (importance scores: 0.164 and 0.126 respectively), providing valuable indicators for the model's predictions. An increase in the values of these features is strongly associated with a respective increase in the value that was predicted at the Level variable. These features played a significant role in influencing the accuracy of the forecasts. However, the Volume48, SumRain7days, SumRain48, and Duration features demonstrated relatively weaker importance, with importance scores of 0.093, 0.060, 0.012, and 0.011 respectively. Although they had a smaller

influence on the model's predictions, they still contributed to the overall performance. In contrast, the Intensity and Rain features exhibited small negative feature importance values. The Intensity feature showed a minimal negative importance ($-0.008$), suggesting a weak inverse relationship with the target variable. Similarly, the Rain feature obtained a negative importance score of $-0.013$, indicating a limited impact on the model's predictive performance. An increase in the values of these features is associated with a slight decrease in the predicted value of the target variable. It should be noted that the influence of these features was relatively minor but in the opposite direction. These findings highlighted the importance of specific features, such as Outflow, MaxLevel48, DryPeriod, and Max48HrRain, in accurately predicting the Level variable. Understanding the impact and relevance of these features enhances the interpretability and effectiveness of the model.

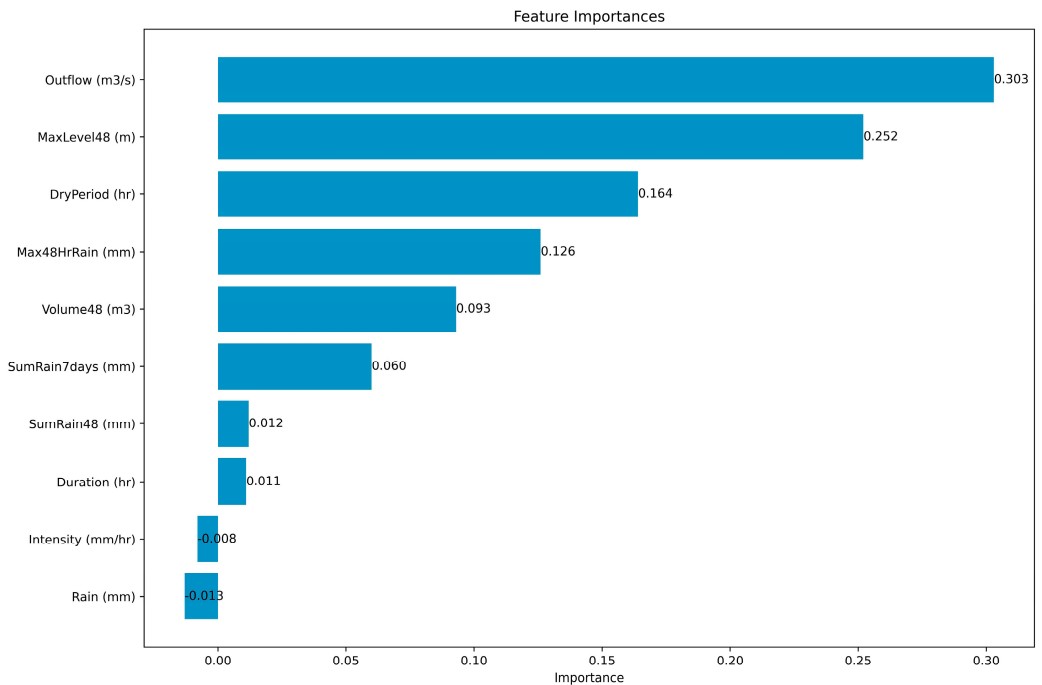

**Figure 7.** Permutation importance analysis results for the 10 additional investigated features.

Figure 8 presents a graphical representation of the feature investigation framework. The evaluation of the forecast results in the three individual stages was based on the Root Mean Squared Error (RMSE) statistical indicator, chosen to compare the impact of features on the forecasting performance. The comparison between the first and second stages revealed an enhancement in the model's effectiveness with the incorporation of additional features. This enhancement was evident as the RMSE value decreased, indicating a reduction in the prediction error. The inclusion of the full set of investigated features contributed to a more accurate and refined forecasting outcome. Furthermore, the comparison across all three stages consistently demonstrated a progressive decrease in the RMSE, indicating a notable improvement in the forecasting performance. In the third stage, the model utilized the top five most important features as inputs. Interestingly, removing the less significant descriptors resulted in a slight improvement in the model's fit. For example, when using 11 descriptors, the RMSE fit score for a 15-min prediction time was 0.008, whereas utilizing only the top five most significant features yielded a score of 0.007. Similarly, for a 60-min prediction time, the RMSE score was 0.0111 with 11 descriptors, compared to 0.0110 with the top five most significant features. Although the decrease in RMSE was modest, it was accompanied by a reduction in computational time, which was a noteworthy achievement. Overall, the graphical visualization highlighted the positive impact of incorporating important features on the forecasting performance, as evidenced by the reduced RMSE values across the three stages. This improvement signified the effectiveness and potential of the

feature investigation framework in enhancing the precision and reliability of the LSTM model's forecasts.

**Figure 8.** RMSE variations for the three stages of feature importance analysis.

### 3.3. LSTM Architecture Performance Using Evaluation Metrics

The five LSTM models' predictions were assessed by a set of performance indices. RMSE measured the discrepancy between observed and simulated values, spanning from 0 to infinity. It is a measure that quantifies the average difference between corresponding model outputs and observations, calculated as the square root of the mean of the squared deviations. Root Mean Squared Logarithmic Error (RMSLE) is a modified version of RMSE that calculates the logarithmic difference between predicted and observed values, reducing the impact of large errors when the observed values exceed model outputs [67]. Mean Absolute Error (MAE) quantifies the average magnitude of the discrepancy between predicted and observed values, normalized by the total number of examples. RMSE and MAE are commonly used assessment metrics for regression tasks, with MAE being more suitable for data with outliers, while RMSE is preferable for comparing different regression models [14]. Lower values of RMSE, RMSLE, and MAE indicate higher forecast accuracy. Conversely, the coefficient of determination ($R^2$), also known as the goodness of fit, should approach one (1) for optimal prediction results, reflecting a robust correlation between the model's outputs and the observed data.

Table 7 provides a comprehensive overview of the evaluation metric results obtained from the multi-step predictions generated by the five LSTM models. These models were trained and validated using the identified hyperparameter values. The table showcases the performance of each model based on the examined statistical indicators. All five LSTM model architectures demonstrated satisfactory performance, as they consistently yielded very good results across all evaluation indicators.

**Table 7.** Statistical indicators for the multi-step predictions of the five examined LSTM models.

| Root Mean Squared Error (RMSE) | 15 min | 30 min | 45 min | 60 min |
|---|---|---|---|---|
| Vanilla LSTM | 0.0073 | 0.0085 | 0.0096 | 0.0109 |
| Stacked LSTM | 0.0073 | 0.0086 | 0.0101 | 0.0113 |
| Bidirectional LSTM | 0.0075 | 0.0087 | 0.0093 | 0.0101 |
| Encoder–Decoder LSTM | 0.0073 | 0.0087 | 0.0100 | 0.0110 |
| Encoder–Decoder Bi-LSTM | 0.0073 | 0.0086 | 0.0097 | 0.0108 |
| Coefficient of Determination ($R^2$) | 15 min | 30 min | 45 min | 60 min |
| Vanilla LSTM | 0.9654 | 0.9541 | 0.9404 | 0.9231 |
| Stacked LSTM | 0.9657 | 0.9531 | 0.9347 | 0.9187 |
| Bidirectional LSTM | 0.9635 | 0.9516 | 0.9439 | 0.9347 |
| Encoder–Decoder LSTM | 0.9661 | 0.9517 | 0.9361 | 0.9217 |
| Encoder–Decoder Bi-LSTM | 0.9654 | 0.9525 | 0.9393 | 0.9244 |
| Root Mean Squared Logarithmic Error (RMSLE) | 15 min | 30 min | 45 min | 60 min |
| Vanilla LSTM | 0.0068 | 0.0075 | 0.0084 | 0.0093 |
| Stacked LSTM | 0.0068 | 0.0076 | 0.0086 | 0.0094 |
| Bidirectional LSTM | 0.0071 | 0.0080 | 0.0084 | 0.0089 |
| Encoder–Decoder LSTM | 0.0068 | 0.0078 | 0.0088 | 0.0096 |
| Encoder–Decoder Bi-LSTM | 0.0069 | 0.0078 | 0.0086 | 0.0093 |
| Mean Absolute Error (MAE) | 15 min | 30 min | 45 min | 60 min |
| Vanilla LSTM | 0.0033 | 0.0034 | 0.0037 | 0.0039 |
| Stacked LSTM | 0.0032 | 0.0035 | 0.0037 | 0.0039 |
| Bidirectional LSTM | 0.0033 | 0.0037 | 0.0039 | 0.0041 |
| Encoder–Decoder LSTM | 0.0032 | 0.0037 | 0.0040 | 0.0043 |
| Encoder–Decoder Bi-LSTM | 0.0033 | 0.0036 | 0.0038 | 0.0040 |

Furthermore, Figure 9 offers a graphical representation of these results through box plots. These box plots illustrate the variations in the multi-step predictions across the examined statistical indicators for all the LSTM architectures. The box plots provide a visual summary of the performance distribution and allow for easy comparison and analysis of the model's predictive capabilities. Together, Table 7 and Figure 9 present a comprehensive assessment of the LSTM models' performance in generating multi-step predictions. These evaluation metrics and visual representations offer valuable insights into the relative strengths and weaknesses of each model, aiding in the selection and interpretation of the most suitable LSTM architecture for the forecasting task.

Upon analyzing the results, it became evident that the Bidirectional LSTM architecture proved to be the most suitable choice for the given datasets. This conclusion was supported by the variations observed in the RMSE and RMSLE indicators, which exhibited smaller deviations among the four different time step predictions and achieved the lowest mean values compared to the other architectures. These indicators served as important metrics to assess the precision of the predictions, and the Bidirectional LSTM architecture demonstrated superior performance in this regard. Additionally, when considering the coefficient of determination, it was notable that the Bidirectional LSTM architecture yielded the highest mean value. This coefficient provided an indication of the degree to which the predicted values corresponded to the observed data, and the larger mean value obtained by the Bidirectional LSTM architecture reinforced its effectiveness in capturing the underlying patterns and trends in the dataset. However, the Stacked and Encoder–Decoder LSTM architectures exhibited poorer results compared to the other LSTM architectures. These two models displayed significant variations between the four time step predictions,

accompanied by higher mean values of RMSE and RMSLE. Additionally, their coefficient of determination values were comparatively lower.

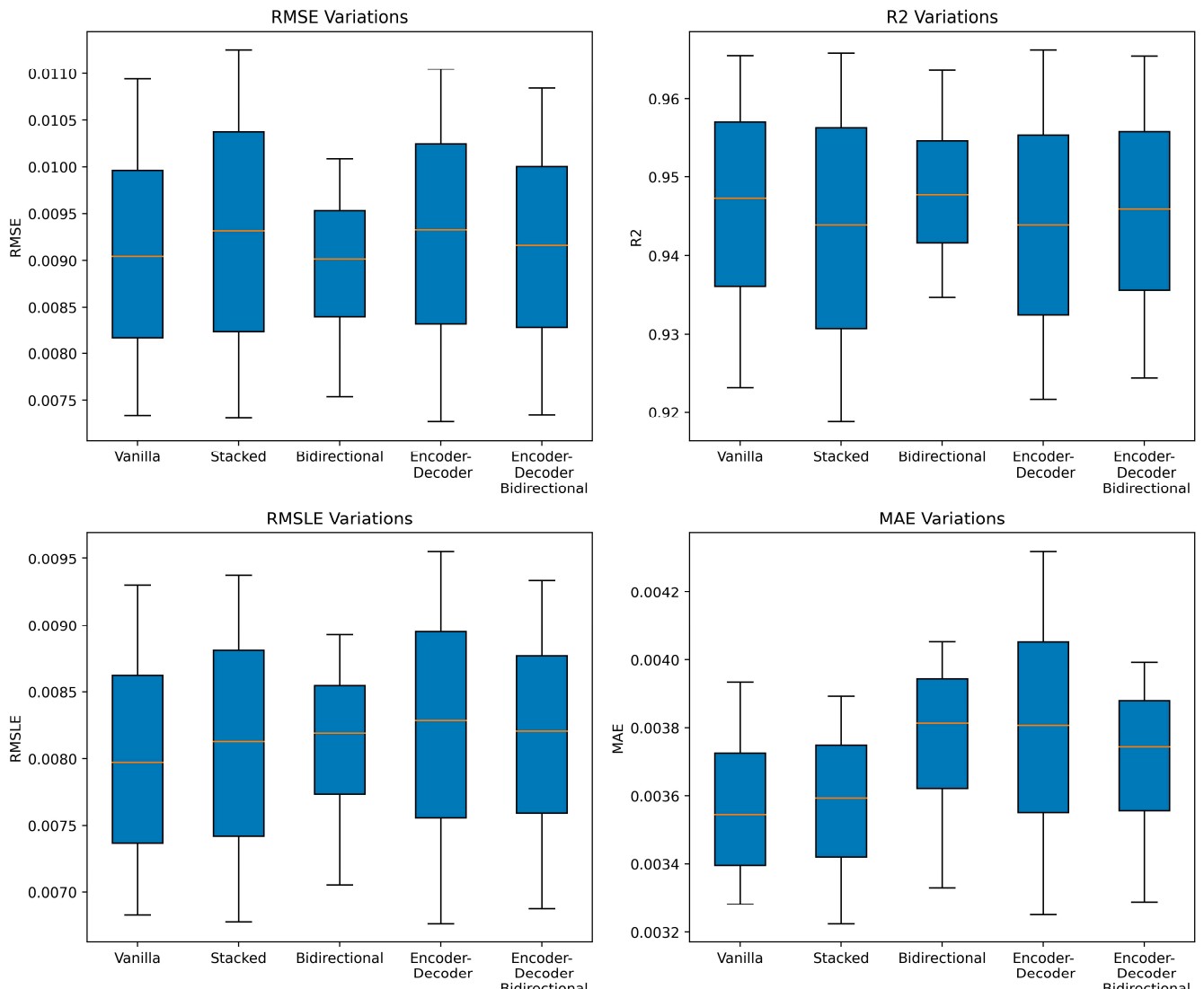

**Figure 9.** Statistical indicators variations for the five examined LSTM model forecasting results.

However, it is worth noting that the evaluation based on the MAE indicator presented a different perspective. In this case, the Vanilla architecture emerged as the preferable choice, as it exhibited more favorable results compared to the other architectures in terms of mean values. Overall, the assessment of these evaluation metrics highlighted the Bidirectional LSTM architecture as the optimal choice for the given datasets, emphasizing its superior performance in terms of RMSE, RMSLE, and $R^2$. Nonetheless, the preference for the Vanilla architecture based on the MAE indicator suggested the presence of varying considerations when evaluating different performance aspects of the models.

### 3.4. Level Multi-Step Predictions

The preferred architecture, Bidirectional LSTM, incorporating the five most significant features as inputs, was employed to visualize the multi-step predictions. A graphical representation of the model's performance is presented in Figure 10, showcasing a comparison between the observed ground truth Level data and the corresponding predicted values across the entire investigation period spanning from November 2018 to November 2022. The visual analysis of the four examined time step predictions generated by the

Bidirectional LSTM model demonstrated a strong concurrence between the observed and predicted datasets. This observation underscored the model's capacity to precisely capture and forecast the dynamics of the system across different time steps.

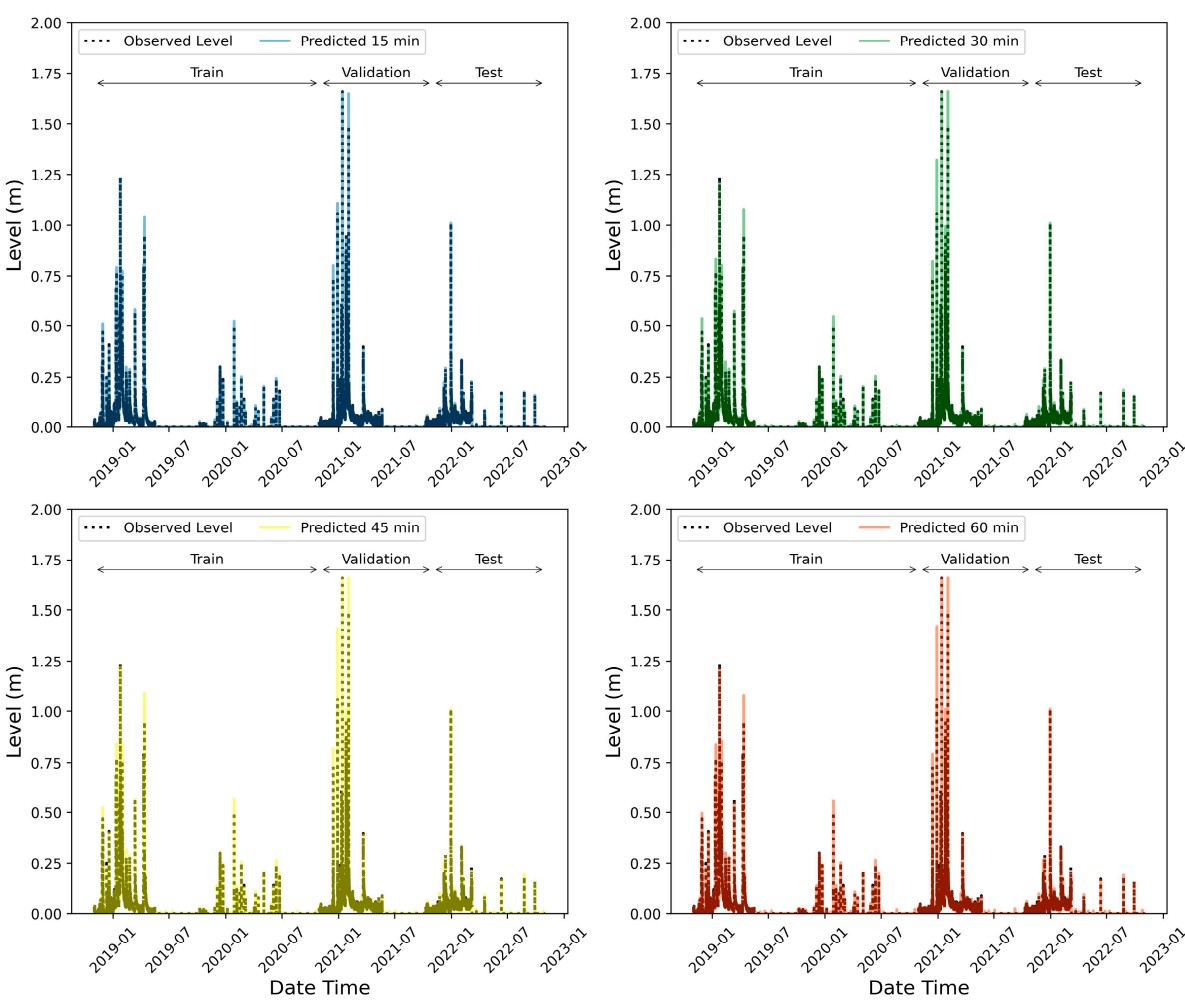

**Figure 10.** Graphical comparison between observed and predicted time series for 15-min, 30-min, 45-min, and 60-min Level prediction using Bidirectional LSTM architecture.

Moreover, Figure 11 displays scatter plots with colored bars, providing a detailed examination of residuals for the four time step forecasts during the testing period spanning from November 2021 to November 2022. These results provided empirical evidence for the reasonable concordance between the observed and simulated Level at a 15-min time scale, further substantiating the effectiveness of the model. Additionally, this graph contributed to a deeper understanding of the deviations observed among the forecasts for the four different time steps. The model demonstrated superior performance at all time steps, including 15 min, 30 min, 45 min, and 60 min. However, it is important to note that the effectiveness of the predictions diminished as the forecasting horizon extended to longer-term predictions. This trend became apparent when comparing the trend line in the 60-min prediction graph, which deviated from the 1:1 line, with the smaller time step predictions.

Figure 12 complements the analysis by visually presenting the differences between the predicted and observed values across the four time steps using violin graphs. Violin plots, akin to box plots, provide a statistical graphic that facilitates the comparison of distributions. This visualization further supported the observation that the model performed best for shorter time step predictions, as evidenced by the narrower and more concentrated distributions in the violin graphs.

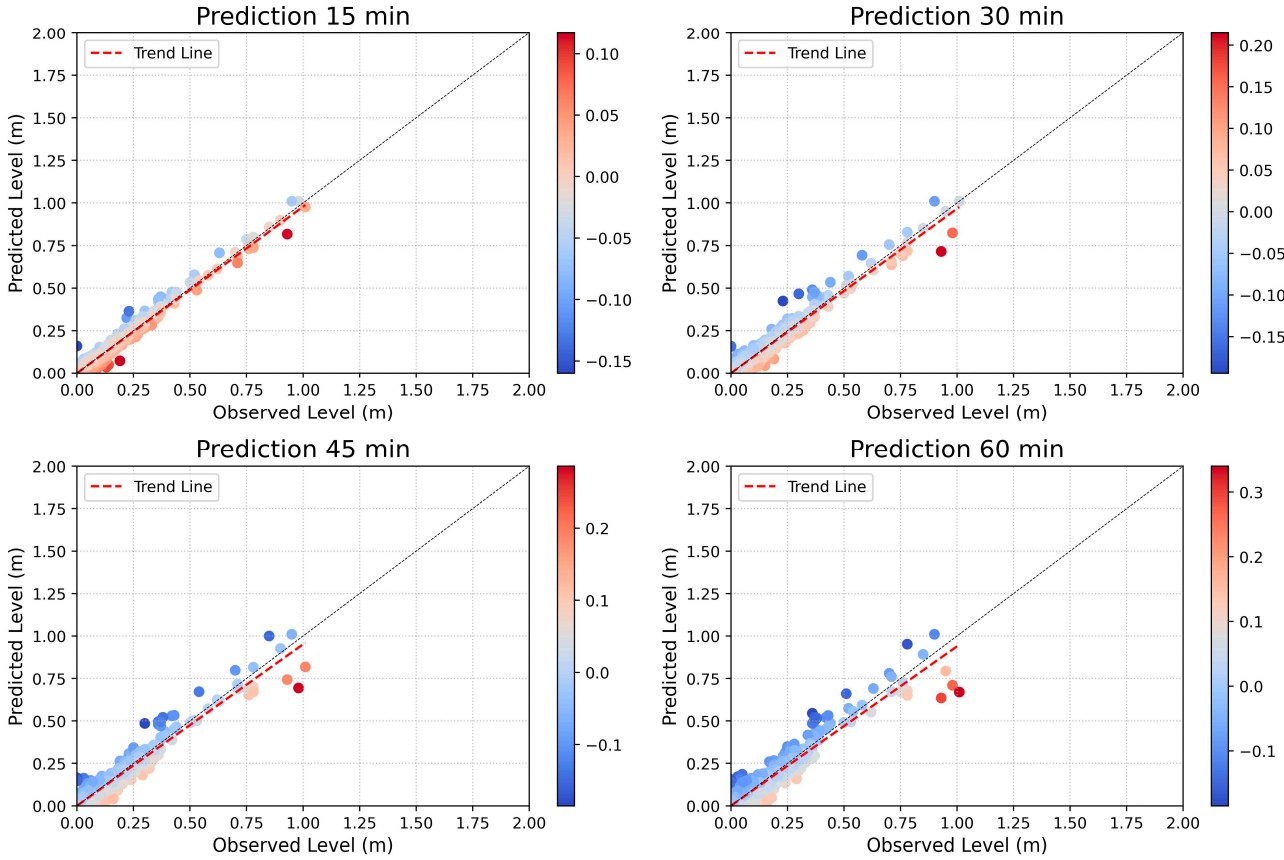

**Figure 11.** Scatter plots with colored bars for residuals, comparing predicted and observed Level values for the four time step forecasts using the Bidirectional LSTM architecture.

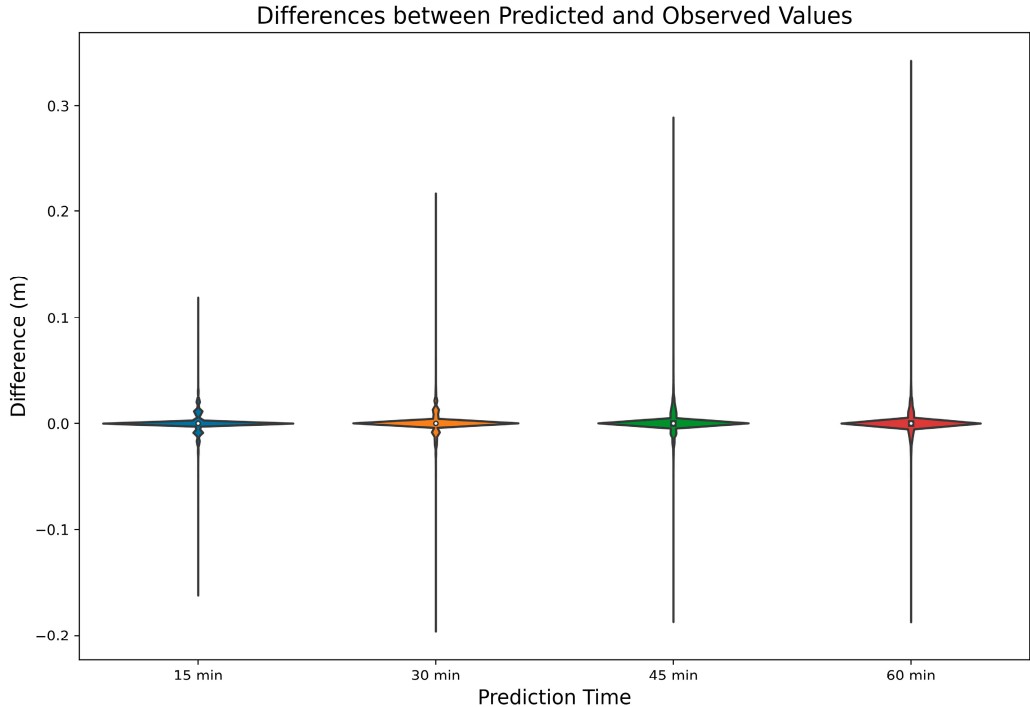

**Figure 12.** Violin plot comparing predicted and observed Level differences in 15 min, 30 min, 45 min, and 60 min ahead forecasts.

## 4. Discussion

The implementation of an early flood monitoring and forecasting system (EMFS) plays a crucial role as a non-structural adaptation measure aimed at enhancing resilience to floods. An EMFS attempts to extract advanced predictions of critical overflow levels, enabling timely alerts to be disseminated to the public and relevant authorities, thereby facilitating preparedness for flash floods. The reliability and effectiveness of such a system depend on various factors, with the quality of input data, including hydrological and meteorological monitoring data from telemetry stations, the performance of the physical hydrological model employed, and the accuracy and reliability of the forecasting process, being key components. The final step involves the effective dissemination of relevant information to end users, ensuring the timely and reliable provision of information to mitigate the potential impacts of flooding events [32]. These constituent elements collectively contribute to the overall quality and efficacy of the flood monitoring and forecasting system, providing timely and reliable information to mitigate the adverse effects of flooding.

Flood prediction is an important aspect of flood management, as it allows for the issuance of early warning alerts and the implementation of necessary emergency measures. The flood prediction models play a crucial role by offering forecasted data that ranges from short-term to long-term, depending on the specific requirements [32]. Advanced machine learning techniques, particularly deep learning models such as artificial neural networks (ANNs) and LSTM architectures have demonstrated significant potential in improving flood predictions. Madaeni et al. [89] compared deep learning techniques (convolutional neural networks (CNN), long short-term memory (LSTM), and combined CNN-LSTM networks) with machine learning methods, for predicting ice jams in rivers in Quebec, Canada. The results indicate that the combination of CNN and LSTM in the CNN-LSTM model yielded the best performance, highlighting the complementary nature of these two deep learning approaches. Zhang et al. [90] compared LSTM, Convolutional Neural Network LSTM (CNN-LSTM), Convolutional LSTM (ConvLSTM), and Spatio-Temporal Attention LSTM (STA-LSTM) models for flood forecasting in the Humber River, Toronto, with the STA-LSTM model outperforming others for forecast times longer than 6 h. Similarly, Xu et al. [17], Ibrahim et al. [91], and Kilsdonk et al. [92] utilized LSTM networks for flood process prediction in the Yellow River watershed, daily discharge forecasting in the Mosul region of Iraq, and flood time series prediction in Hooglanderveen, the Netherlands, respectively. Barrera-Animas et al. [67] compared Vanilla LSTM, Stacked LSTM, and Bidirectional LSTM networks for rainfall forecasting in the UK, highlighting the comparable performance of the Bidirectional LSTM network.

Accurately forecasting time series data, specifically water levels for flood-warning systems, poses a crucial challenge due to the complex linear and nonlinear correlation structures inherent in water-stage time series [65]. To address this, it is imperative to employ time series hydrological prediction models that can unveil hidden information and provide reliable predictions for effective flood management [65]. Data-driven LSTM architectures, integrated with real-time monitoring systems, have demonstrated remarkable potential in enhancing the prediction and management of urban floods recently. In a comparative study conducted by Atashi et al. [65], hourly floodwater level prediction in the basin of the Red River of the North, Canada, was evaluated using three different methods: a classical statistical method (SARIMA), a classical machine learning algorithm (Random Forest), and a state-of-the-art deep learning method (LSTM). The LSTM method exhibited superior performance compared to SARIMA and Random Forest, leveraging real-time monitoring data from three water level gauge stations and three flow-gauging stations. Similarly, Gohar et al. [14] compared the LSTM model with RNN using Backpropagation Through Time (BPTT) for river level prediction at Hoppers Crossing station, Melbourne, Australia, across various time intervals ranging from 1 to 12 h in advance. The study highlighted the potential of LSTM for short-term predictions while noting its limitations in capturing peak values for 8 and 12 h predictions. Furthermore, Noor et al. [93] developed LSTM and attention-based architectures to forecast daily flood water levels in Bangladeshi rivers, with the spatial and

temporal attention LSTM (STALSTM) model demonstrating superior performance over other attention-based LSTM models. This study emphasized the significance of accurate flood predictions enabled by the STALSTM-based system in informing flood management plans in Bangladesh and beyond. Additionally, Won et al. [12] investigated deep learning models, including Vanilla ANN, LSTM, Stack LSTM, and Bidirectional LSTM, utilizing 10-min hydrological time series data from monitoring stations to forecast flood risks based on water levels in the Dorim river basin, Seoul, South Korea. These deep learning models, trained on high-resolution hydrological data, were effective in providing timely warnings for anticipated flood risks in urban streams.

Enhancing understanding and management of flood risk in urban areas necessitates a consideration of the unique hydrological characteristics exhibited by urban streams, particularly their intermittent flow patterns. An intriguing avenue to improve real-time forecasts is through the application of data-driven techniques that augment deterministic lumped rainfall-runoff models, wherein catchment response is simulated using physically-based models like HEC-HMS. While numerous studies have compared the prediction outcomes of data-driven and physical models in hydrology, as evidenced by Rauf and Ghumman [94], Hu et al. [95], Abbas et al. [96], and Cai and Yu [38], the integration of these two approaches remains relatively scarce, with only a few notable examples such as the work of Won et al. [12].

Notwithstanding the valuable insights gained from recent research, there remains a significant gap in exploring the application of advanced deep learning models in conjunction with real-time monitoring systems for predicting multi-step river water levels, particularly in small-scale urban basins characterized by rapid response times. Additionally, the majority of existing studies have focused on daily and hourly time steps for input data. Notably, none of the studies mentioned in these reviews have presented a data-driven model that integrates real-time monitoring and a physically-based hydrological model with sub-hourly time steps for the simultaneous multi-step prediction of water levels provided. Addressing this critical research area, the present study investigated five different state-of-the-art LSTM architectures, along with their optimal hyperparameters. The LSTM water level predictions were seamlessly integrated by incorporating the outputs of hydrological variables from the physically-based HEC-HMS model and using real-time data from telemetry monitoring systems. The current study leveraged the Long Short-Term Memory (LSTM) method, an advanced Deep Learning technique that has undergone comprehensive investigation and demonstrated remarkable efficacy in forecasting hydrological time series. Notably, the LSTM model showcased its competence in accurately capturing both the linear and nonlinear correlation structures inherent in water-stage time series data [65].

Time series datasets with high temporal sub-hourly frequency were utilized as inputs, specifically tailored to predict water levels for four time horizons ranging from 15 min to 60 min. The multi-time step predictions showed a decline in effectiveness as the forecasting horizon extended to longer-term predictions. This aligns with the study by Chen et al. [13] that found a decrease in the accuracy of flow prediction as the time for predicting future flow increases. Adjusting expectations for longer-term forecasts is essential due to the diminishing effectiveness of predictions. This approach is designed to cater to the needs of small-scale streams with very small response times, emphasizing the importance of accurately forecasting water levels in such scenarios.

The models' results can be sensitive to the choice of feature representation, model hyperparameters, and evaluation metrics. For this reason, a permutation feature importance investigation and grid search hyperparameter tuning were conducted, along with the computation of four different evaluation metrics, aiming to assess the models' performance accuracy. While the Permutation Feature Importance algorithm showed promise in understanding input–output relationships in neural networks, its application in hydrologic studies remains limited [45]. The obtained feature importance scores provide valuable insights into both the data and the model, aiding in dimensionality reduction and improving model performance. By employing permutation feature importance for feature

selection, this study aimed to identify the most influential variables for accurate forecasting using deep learning algorithms. The results highlight the significance of attributes such as Outflow, MaxLevel48, DryPeriod, and Max48HrRain, which positively contribute to the accuracy of the LSTM model's predictions. Conversely, attributes such as Rain and Intensity had a relatively minor negative impact on the model's predictions. These findings suggest that the LSTM models prioritize attributes derived directly from the available time series data, indicating potential for improving the long-term memory of the machine learning models. Understanding the relative importance of these features provides valuable information about the factors that influence the model's performance, and enhances the interpretability of the LSTM model.

The results obtained from the examined LSTM architectures highlight their high precision and forecasting capability for floodwater levels across different prediction time frames. All the evaluated LSTM architectures demonstrated accurate predictions based on the evaluation statistical metrics. The RMSE, which was much smaller than the scale of the data, indicated that the models performed well in terms of prediction accuracy. This suggests that the models effectively capture the patterns and fluctuations in the time series data, resulting in predictions that closely align with the original values. Different evaluation metrics were employed to comprehensively assess the model's performance during training and after making predictions, providing varied perspectives on its effectiveness. Among the five examined LSTM architectures, the Bidirectional LSTM consistently yielded superior prediction results, particularly when considering the mean of the four-time horizons. The Vanilla LSTM architecture also showed satisfactory results, performing best according to the MAE indicator. Encoder–Decoder architectures, particularly the Encoder–Decoder Bi-LSTM, exhibited promising outcomes for longer-term predictions, such as the 1-h horizon. However, after considering the overall performance across different time horizons and statistical indicators, the simple Bidirectional LSTM emerged as the optimal choice for the current dataset. This finding aligns with previous studies that have demonstrated the superiority of the Bidirectional LSTM architecture in rainfall forecasting [67] and runoff prediction tasks [68]. These studies highlight the competitive performance of Bidirectional LSTM, particularly when equipped with a sequence-to-sequence architecture. The generality of the model structure is evident from its successful application to different river basins, further supporting its efficacy.

## 5. Conclusions

This study has introduced an advanced system called the Early Flood Monitoring and Forecasting System (EMFS), designed specifically for predicting critical overflow levels in a small-scale urban stream. Given the region's history of severe flash flood incidents that demand a swift response, this research aimed to investigate the influence of different sub-hourly input sequences on various LSTM architectures. Through the proposal of a sensitive LSTM architecture and optimization of the input sequence, the study has achieved accurate multi-step prediction of water levels, yielding valuable insights and conclusions. The LSTM models were trained and validated using multivariate time series data, focusing on prediction times ranging from 15 min to 1 h, addressing the specific response needs of the basins. The graphical representations demonstrate the reliability and accuracy of the Bidirectional LSTM model in predicting water levels. The close agreement between the observed and predicted datasets, particularly at the 15-min time scale, showcases the model's ability to capture the system's dynamics effectively. The developed EMFS serves as a non-structural approach to mitigate urban flood damage and can be further extended to other locations, taking into account specific watershed characteristics.

**Author Contributions:** Conceptualization, Eleni-Ioanna Koutsovili and Ourania Tzoraki; methodology, Eleni-Ioanna Koutsovili; software, Eleni-Ioanna Koutsovili; validation, Nicolaos Theodossiou and George E. Tsekouras; formal analysis, Eleni-Ioanna Koutsovili; investigation, Eleni-Ioanna Koutsovili; resources, Ourania Tzoraki; data curation, Eleni-Ioanna Koutsovili; writing—original draft preparation, Eleni-Ioanna Koutsovili; writing—review & editing, George E. Tsekouras and Nicolaos Theodossiou; visualization, Eleni-Ioanna Koutsovili; supervision, Ourania Tzoraki and Nicolaos Theodossiou; project administration, Ourania Tzoraki; funding acquisition, George E. Tsekouras. All authors have read and agreed to the published version of the manuscript.

**Funding:** This research was part of E.I. Koutsovili's doctoral thesis. The implementation of the doctoral thesis was co-financed by Greece and the European Union (European Social Fund-ESF) through the Operational Programme «Human Resources Development, Education and Lifelong Learning» in the context of the Act "Enhancing Human Resources Research Potential by undertaking a Doctoral Research" Sub-action 2: IKY Scholarship Programme for PhD candidates in the Greek Universities». Additionally, the author O. Tzoraki is funded by the Operational Program National Strategic Reference Framework (NSRF) North Aegean 2014–2020, project name "Observatory of Coastal Environment –AEGIS+".

**Data Availability Statement:** The data presented in this study are available on request from the corresponding author.

**Conflicts of Interest:** The authors declare no conflict of interest. The funders had no role in the design of the study; in the collection, analyses, or interpretation of data; in the writing of the manuscript; or in the decision to publish the results.

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
