# Peer review of "Early Flood Monitoring and Forecasting System Using a Hybrid Machine Learning-Based Approach"

_ijgi, doi:10.3390/ijgi12110464_

Round 1
Reviewer 1 Report
Comments and Suggestions for Authors
1. The author should further explain what is HEC-HMC at line 18 since the readers do not understand the abbreviation.
2. The paper employed EMFS to predict the critical overflow level and used various parameters and data. I want to know which criteria are key to the predicted results. Meanwhile, the slope has a vital influence on the flood, how to evaluate its performance in your experiment? Please see and cite the following papers [1,2].
3. The paper demonstrates EMFS can produce higher accuracy for Lesvos Island, Greece. Does the system bring in the same higher results in other catchments?
4. It is better to add some literature to demonstrate Hec-HMS plays a pivotal role in elaborating hydrological time-series data at line 135.
5. The paper used DEM, Land use, and soil maps in ArcGIS at line 140 and data in Table 1. Kinds of data have different resolutions. The author should detail the properties of these datasets, and I also wondered how differing datasets were integrated into the proposed framework with the same resolution.
6. The author should detail why used 4 hours to make forecasts at line 162. Why not another time?
7. The maps in the up-right corner and the bottom-right corner of Figure 2 do not have legends and scales, and the annotation is too small.
8. The experiment chose 50% training, 25% validation, and 25% test subsets from lines 242 to 244. There are two problems, one is how to determine these ratios in the study or should explain. The other is the trained model may be just for the fixed 25% test subsets based on 50% training, but if the test subsets changed, the model accuracy may be decreased. The author should discuss various ratios that have different effects on the accuracy of the model.
9. Potential maximum retention is determined by Soil Water Retention (SWR) and CN values are determined by the units used, e.g. SWR = 10 for units of inches, and SWR = 254 for units of millimeters. The paper should brief the SWR value in the experiment (SWR = 10).
10. Sensitivity Analysis should be conducted further to explore what’s relationship between the inputs and the outputs of the model, which can help the authors make further investigations in the future.
11. Some sentences are better to adjust their order, making their logic smoother. At line 210, Two additional meteorological stations in Stypsi and Agia Paraskevi have been operational since 2018 and 2003 respectively, revised to Two additional meteorological stations in Agia Paraskevi and Stypsi have been operational since 2003 and 2018 respectively.
12. The format of the references should be consistent. For example, The 8th reference does not have pp at the end, but the 9th reference has one.
Comments on the Quality of English Language
English of the paper needs to be proofread by native speakers, such as In the literature, there exist various approaches for flood forecasting… at line 69.
Author Response
Thank you very much for taking the time to review this manuscript. We are grateful for your valuable feedback.
Please find our detailed responses to each of your comments in the attached document.

Reviewer 2 Report
Comments and Suggestions for Authors
All comments are in the attached file

Author Response
Thank you for dedicating your time to review our manuscript. We sincerely appreciate your valuable feedback.
Please find our detailed responses to each of your comments in the attached document.

Reviewer 3 Report
Comments and Suggestions for Authors
References need to be arranged alphabetically
Author Response
Thank you very much for taking the time to review this manuscript. We appreciate your valuable feedback.
Regarding your suggestion for references formatting, we have adhered to the guidelines provided by the journal for arranging the reference list.
Round 2
Reviewer 1 Report
Comments and Suggestions for Authors
No further suggestions.
Comments on the Quality of English LanguageEnglish is fine, no further suggestions.